# Ribozyme-catalysed RNA synthesis using triplet building blocks

James Attwater, Aditya Raguram[†], Alexey S Morgunov, Edoardo Gianni, Philipp Holliger*

MRC Laboratory of Molecular Biology, Cambridge Biomedical Campus, Cambridge, United Kingdom

**Abstract** RNA-catalyzed RNA replication is widely believed to have supported a primordial biology. However, RNA catalysis is dependent upon RNA folding, and this yields structures that can block replication of such RNAs. To address this apparent paradox, we have re-examined the building blocks used for RNA replication. We report RNA-catalysed RNA synthesis on structured templates when using trinucleotide triphosphates (triplets) as substrates, catalysed by a general and accurate triplet polymerase ribozyme that emerged from in vitro evolution as a mutualistic RNA heterodimer. The triplets cooperatively invaded and unraveled even highly stable RNA secondary structures, and support non-canonical primer-free and bidirectional modes of RNA synthesis and replication. Triplet substrates thus resolve a central incongruity of RNA replication, and here allow the ribozyme to synthesise its own catalytic subunit '+' and '−' strands in segments and assemble them into a new active ribozyme.

DOI: https://doi.org/10.7554/eLife.35255.001

*For correspondence:
ph1@mrc-lmb.cam.ac.uk

Present address: †Department of Chemistry and Chemical Biology, Harvard University, Cambridge, United States

Competing interests: The authors declare that no competing interests exist.

## Introduction

The premise that some RNA sequences can catalyse and template their own replication - reciprocally synthesizing their own '+' and '−' strands - underpins current thinking about early genetic systems (*Crick, 1968*; *Orgel, 1968*; *Szostak et al., 2001*). Any ancient ribozyme with such RNA replicase capability seems to be lost, but efforts are ongoing to recreate RNA self-replication in the laboratory (*Martin et al., 2015*) as a critical test of the 'RNA world' hypothesis (*Gilbert, 1986*). Early on, derivatives of naturally occurring self-splicing introns (*Doudna et al., 1991*; *Green and Szostak, 1992*; *Hayden and Lehman, 2006*) as well as later in vitro evolved ligase ribozymes (*Lincoln and Joyce, 2009*; *Sczepanski and Joyce, 2014*) were shown to be able to assemble one of their own strands from cognate constituent RNA segments. However, a critical drawback of such systems is their need for specific preformed building blocks of at least eight nucleotides (nt) average length, limiting their potential for open-ended evolution, and precluding their replication from pools of random-sequence oligonucleotide substrates (*Green and Szostak, 1992*; *Doudna et al., 1993*).

In a contrasting approach, RNA polymerase ribozymes (RPRs) have been developed that can use general monomer building blocks (ribonucleoside 5' triphosphates (NTPs)) in RNA-templated RNA synthesis (*Johnston et al., 2001*; *Zaher and Unrau, 2007*; *Wochner et al., 2011*; *Attwater et al., 2013b*; *Horning and Joyce, 2016*), akin to the activity of modern proteinaceous polymerases. However, even the most highly-evolved RPRs (*Horning and Joyce, 2016*) are substantially impeded by template secondary structures. Such structures are ubiquitous in larger, functional RNAs (including the RPRs themselves) and generally indispensable for function. The strong inhibitory role of this central feature of RNA leads to an antagonism between the degree to which an RNA sequence is able to fold into a defined three-dimensional structure to encode function (such as catalysis) and the ease with which it can be replicated (*Boza et al., 2014*). This ostensible 'structure vs. replication' paradox

**eLife digest** Life as we know it relies on three types of molecules: DNA, which stores genetic information; proteins that carry out the chemical reactions necessary for life; and RNA, which relays information between the two. However, some scientists think that before life adopted DNA and proteins, it relied primarily on RNA.

Like DNA, strands of RNA contain genetic data. Yet, some RNA strands can also fold to form ribozymes, 3D structures that could have guided life's chemical processes the way proteins do now. For early life to be built on RNA, though, this molecule must have had the ability to make copies of itself.

This duplication is a chemical reaction that could be driven by an 'RNA replicase' ribozyme. RNA strands are made of four different letters attached to each other in a specific order. When RNA is copied, one strand acts as a template, and a replicase ribozyme would accurately guide which letters are added to the strand under construction. However, no replicase ribozyme has been observed in existing life forms; this has led scientists to try to artificially create RNA replicase ribozymes that could copy themselves.

Until now, the best approaches have assumed that a replicase would add building blocks formed of a single letter one by one to grow a new strand. Yet, although ribozymes can be made to copy straight RNA templates this way, folded RNA templates – including the replicase ribozyme itself – impede copying. In this apparent paradox, a ribozyme needs to fold to copy RNA, but when folded, is itself copied poorly. Here, Attwater et al. wondered if choosing different building blocks might overcome this contradiction.

Biochemical techniques were used to engineer a ribozyme that copies RNA strands by adding letters not one-by-one, but three-by-three. Using three-letter 'triplet' building blocks, this new ribozyme can copy various folded RNA strands, including the active part of its own sequence.

This is because triplet building blocks have different, and sometimes unexpected, chemical properties compared to single-letter blocks. For example, these triplets work together to bind tightly to RNA strands and unravel structures that block RNA copying.

All life on Earth today uses a triplet RNA code to make proteins from DNA, and these experiments showed how RNA triplets might have helped RNA sustain early life forms. Further work is now needed to improve the ribozyme designed by Attwater et al. for efficient self-copying.

DOI: https://doi.org/10.7554/eLife.35255.002

would have placed stringent probability constraints on the emergence of an RNA replicase and generally impeded the ability of RNA to function as an early genetic polymer.

We wondered whether this paradox might be avoided through a re-consideration of plausible building blocks for early RNA replication. Models of non-enzymatic polymerisation of all four activated ribonucleotides – the presumed source of the first RNA sequences – yield pools of di-, tri- and tetranucleotide etc. length oligonucleotides (in decreasing abundance) dominating the population alongside longer products (*Monnard et al., 2003*). Here, we have examined whether substrates of such lengths can support RNA-catalyzed RNA replication, by developing a ribozyme capable of iterative templated ligation of 5'-triphosphorylated RNA trinucleotides (henceforth called triplets). This heterodimeric triplet polymerase ribozyme demonstrated a striking capacity to copy a wide range of RNA sequences, including highly structured, previously intractable RNA templates, as well as its own catalytic domain and encoding template in segments. Its characterization revealed emergent properties of triplet-based RNA synthesis, including cooperative invasion and unraveling of stable RNA structures by triplet substrates, bi-directional (both 5'−3' and 3'−5') and primer-free (triplet-initiated) RNA synthesis, and fidelity augmented by systemic properties of the random triplet pools.

## Results

### In vitro evolution of triplet polymerase activity

We set out to explore the potential of short RNA oligonucleotides as substrates for RNA-catalyzed RNA replication. To do this, we required a ribozyme capable of general, iterative RNA-templated

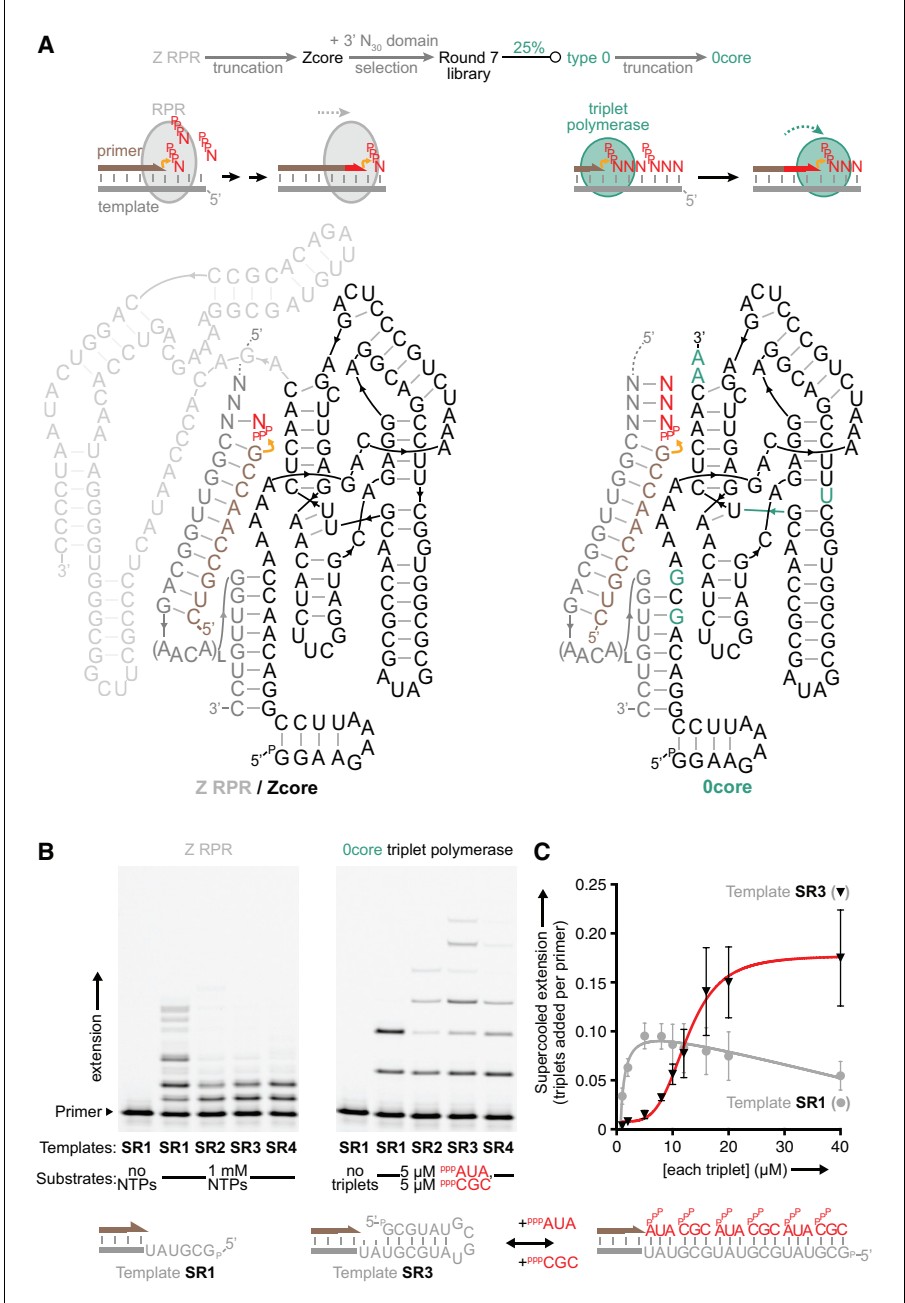

**Figure 1.** Monomer polymerisation and triplet polymerisation. (**A**) Scheme outlining initial derivation of a triplet polymerase activity from a mononucleotide polymerase ribozyme via directed evolution. Z RPR truncation effects are shown in *Figure 1—figure supplement 1*, the selection cycle is outlined in *Figure 1—figure supplement 2*, and the selection conditions of rounds 1–7 are listed in *Figure 1—source data 1*. Below, modes of action and secondary structures of the mononucleotide polymerase ribozyme (Z RPR) and a triplet polymerase ribozyme (0core), both depicted surrounding primer (tan)/template (grey) duplexes with a mononucleoside triphosphate (NTP) or trinucleotide triphosphate (triplet) substrate present (red). Here, the templates are hybridised to the ribozyme upstream of the primer binding site, flexibly tethered to enhance local concentration and activity (via L repeats of an AACA sequence, for example L = 5 in templates SR1-4 below). Z RPR residues comprising its catalytic core (Zcore) are black; mutations in 0core arising from directed evolution of Zcore are in teal. (**B**) Primer extension by the Z RPR using monomers (1 mM NTPs) or by 0core using triplets (5 µM $^{PPP}$AUA and $^{PPP}$CGC), on a series of 6-nucleotide repeat templates (SR1-4, examples below) with escalating secondary structure potential that quenches Z RPR activity beyond the shortest template SR1 (−7°C ice 17 days, 0.5 µM/RNA). Extension by the triplet polymerase ribozyme 0core can overcome these structure tendencies up to the longest template SR4. (**C**) Triplet concentration dependence of extension using templates SR1 (grey circles) and SR3 (black triangles) by 0core ($^{PPP}$AUA and $^{PPP}$CGC, 0.1 µM of primer A10, template and ribozyme, −7°C supercooled 15 days, ± s.d., n = 3); shown below is a model of cooperative triplet-mediated unfolding of template SR3 structure to explain the sigmoidal triplet concentration dependence (red curve) of extension upon it. Numerical values are supplied in *Figure 1—source data 2*.
*Figure 1 continued on next page*

*Figure 1 continued*

DOI: https://doi.org/10.7554/eLife.35255.003

The following source data and figure supplements are available for figure 1:

**Source data 1.** Selection conditions of rounds 1–7.
DOI: https://doi.org/10.7554/eLife.35255.006
**Source data 2.** Triplet concentration-dependent extension values.
DOI: https://doi.org/10.7554/eLife.35255.007
**Figure supplement 1.** Templated ligase activity from a mononucleotide polymerase.
DOI: https://doi.org/10.7554/eLife.35255.004
**Figure supplement 2.** Selection scheme for in vitro evolution of triplet polymerase activity.
DOI: https://doi.org/10.7554/eLife.35255.005

oligonucleotide ligation. Previously-described RNA polymerase ribozymes such as the 'Z' RPR (*Wochner et al., 2011*) can use NTPs to iteratively extend a primer hybridized to an RNA template, but do not accommodate oligonucleotides bound downstream of the primer or accept them as substrates. However, we detected a weak templated ligation activity in a truncated version of the Z RPR comprising its catalytic core domain (Zcore) (*Figure 1a*), which supported incorporation of oligonucleotide substrates as short as three nt (*Figure 1—figure supplement 1*) when incubated in the eutectic phase of water ice (*Attwater et al., 2010*; *Mutschler et al., 2015*).

To be able to properly examine such RNA trinucleotide triphosphates (triplets) as replication substrates, we first sought to convert Zcore into an effective triplet polymerase ribozyme using in vitro evolution. We devised a selection strategy that required iterative templated triplet ligation by ribozymes to achieve their covalent linkage to a tagged primer (*Figure 1—figure supplement 2*). This enables their recovery, amplification and mutagenesis before further rounds of selection to enrich the selection pool in improved triplet polymerase ribozyme variants.

We initiated selections from a library of $1.5 \times 10^{15}$ Zcore variants with a new random 3' $N_{30}$ region under eutectic phase conditions that increase RNA half-life and enhance ribozyme activity (*Attwater et al., 2010;Attwater et al., 2013b*). After 7 rounds of in-ice evolution, one-quarter of the selection pool comprised an improved ribozyme (type 0). Its core domain (0core, *Figure 1a*) could catalyse the iterative polymerization of multiple triplets allowing us to begin to investigate the properties of triplet-based RNA replication.

Significantly, we found that 0core could catalyze triplet polymerisation on a series of structured templates, which had proven intractable to the parental Z RPR (*Figure 1b*). Here, primer extension exhibited a steep sigmoidal dependence upon triplet concentrations (*Figure 1c*), suggestive of a cooperative invasion and unraveling of template secondary structures by the triplet substrates themselves. Although still inefficient, the fact that the nascent activity of the 0core ribozyme could already copy templates that had confounded an established RPR encouraged us to continue to seek improved triplet polymerase ribozymes to leverage this substrate behaviour.

## Emergence of cooperativity and characterisation of a ribozyme heterodimer

We continued selections for a further 14 rounds. At this point, the type 0 ribozyme had gone extinct, replaced by six new types of RNA each characterised by a unique 3' domain (*Figure 2a*, *Figure 2—figure supplement 1*). Type 1 RNAs were the most abundant, comprising ~50% of pool sequences, but mysteriously were catalytically inactive with diverse mutations in their core domains. In contrast, the type 2–6 RNAs all displayed triplet polymerase activity, but fell short of the polyclonal activity of the selection pool (*Figure 2—figure supplement 2*). To attempt to explain this discrepancy, we explored potential interactions among the different pool lineages, and found that addition of an equimolar amount of type 1 RNA substantially enhanced triplet polymerase activity of all the other ribozyme types 2–6 (*Figure 2b*).

Dissecting type 1 RNA function, we found that 5' truncation of the region that previously contacted the primer/template duplex (*Shechner et al., 2009*) did not affect its cofactor activity (*Figure 3a*, *Figure 3—figure supplement 1*). As judged by gel mobility shift (*Figure 3b*) and activity enhancement (*Figure 3—figure supplement 1*), type 1 RNA appears to form a 1:1 heterodimeric

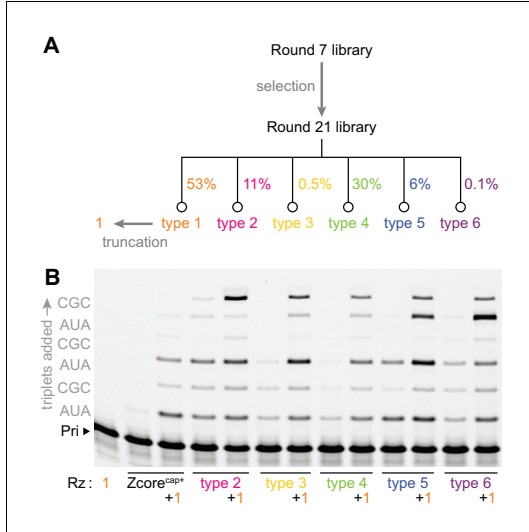

**Figure 2.** Emergence of cooperativity during in vitro evolution. (**A**) Composition of the round 21 selection pool as a % of total pool sequences; selection conditions of rounds 8–21 are listed in *Figure 2—source data 1*. Secondary structures of ribozyme type 1–6 archetypes are shown in *Figure 2—figure supplement 1*, and comparison of their activities to that of the polyclonal selection pool is shown in *Figure 2—figure supplement 2*. (**B**) Primer extension with triplets by these emergent triplet polymerase ribozyme types 2–6 ('Rz', alongside the starting Zcore ribozyme with 'cap+' sequence from selection, see *Figure 3a*), alone or with added truncated type 1 (+1, see *Figure 3a*) which boosted their triplet polymerase activities (0.5 µM Rzs/A10 primer/SR3 template, 5 µM $^{PPP}$AUA and $^{PPP}$CGC, −7°C ice 16 hr).

DOI: https://doi.org/10.7554/eLife.35255.008

The following source data and figure supplements are available for figure 2:

**Source data 1.** Selection conditions of rounds 8–21.
DOI: https://doi.org/10.7554/eLife.35255.011

**Figure supplement 1.** Secondary structures of type 1–6 ribozymes.
DOI: https://doi.org/10.7554/eLife.35255.009

**Figure supplement 2.** Clonal versus polyclonal activity.
DOI: https://doi.org/10.7554/eLife.35255.010

complex directly with active triplet polymerase ribozymes. Our attention was drawn to their selection construct-derived 5' hairpin elements, which differed between active triplet polymerases ('cap+', *Figure 3a*, *Figure 2—figure supplement 1*) and the most common type 1 variants in the selection pool where this hairpin had acquired a mutation (yielding 'cap–', *Figure 3a*). 'cap–' was dispensable for type 1's cofactor activity, but when replacing 'cap+' in active triplet polymerases it abolished both their activity enhancement by type 1 (*Figure 3—figure supplement 1*) and complex formation (*Figure 3b*). This points to the 'cap+' hairpin as the critical site of interaction with type 1; 'cap–' in type 1 presumably served to deter its homodimerisation during selection.

Indeed, transplanting the 'cap+' element could make the parental ribozymes (Zcore and Z RPR) receptive to activity enhancement by type 1 RNA (*Figure 2b*, *Figure 3—figure supplement 2*). The catalytically inert type 1 RNA thus represents a general, mutualistic RNA species. This molecular symbiont appears to have emerged spontaneously during in vitro evolution by forming a heterodimeric holoenzyme with triplet polymerase ribozymes, enhancing their activity to boost the recovery prospects of both complex components.

In complex with type 5 (the fastest enriching triplet polymerase ribozyme in the final selection pool), type 1 boosts polymerization of triplets (or longer oligonucleotides) to enable synthesis of long RNAs (*Figure 3c*). Here, it became apparent that type 1 also obviates the need for ribozyme-template tethering. Due to their poor affinity for primer/template duplex (*Lawrence and Bartel, 2003*), RPRs generally depend upon such tethering to template (*Attwater et al., 2010*; *Wochner et al., 2011*; *Horning and Joyce, 2016*), which enhances local ribozyme concentration and promotes formation of the RPR-primer/template holoenzyme (*Attwater et al., 2010*; *Attwater et al., 2013a*). In contrast, the triplet polymerase heterodimer appears to have a capacity for true intermolecular, sequence-general interaction with primer-template duplexes, which enables holoenzyme formation and copying of RNA templates without requiring specific ribozyme-template hybridization sites.

## Secondary structure invasion by triplet substrates

We performed an additional five rounds of in vitro evolution to further evolve the type 5 triplet polymerase ribozyme (now in the presence of truncated type 1 RNA), diversifying the previously-fixed 3' domain reverse transcription primer binding sequence. This reselection yielded a shorter final heterodimeric triplet polymerase holoenzyme, hereafter termed 't5$^{+1}$' (*Figure 4*). This robust triplet polymerase activity now proved suitable for exploring the scope and potential of triplet-based RNA replication.

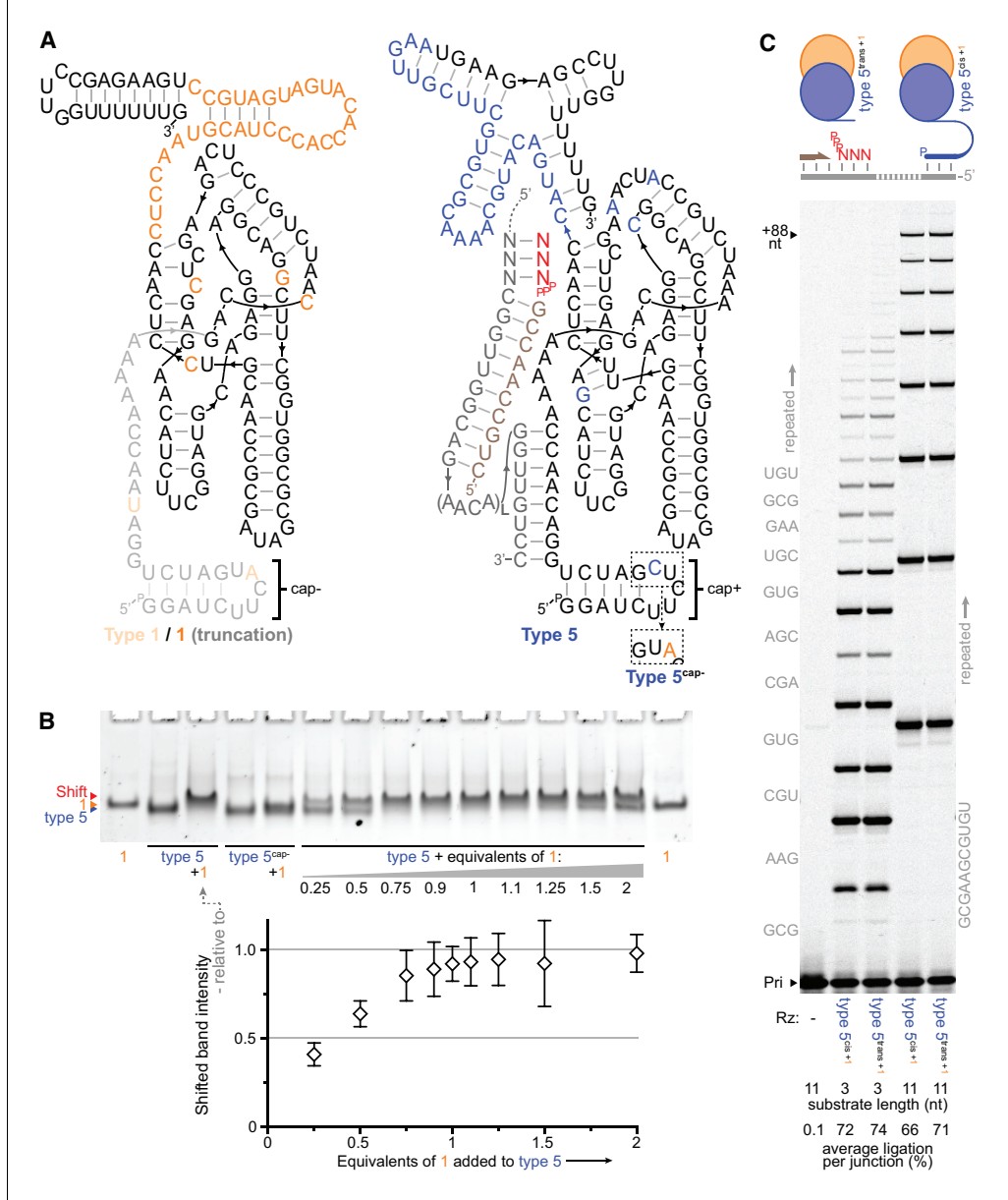

**Figure 3.** Heterodimer formation and behaviour. (**A**) Secondary structures of the most common type 1 and type 5 clones from the selection, with in vitro-selected 3' domain and core mutations coloured orange (type 1) or blue (type 5). 5' truncation of type 1 (faded, including its putative primer/template interacting region), yielding the minimal type 1 variant '1', maintained its enhancement activity (see *Figure 3—figure supplement 1*). The effects of transplanting the indicated 5' hairpin 'cap+' element from type 5 to other ribozymes are shown in *Figure 3—figure supplement 2*. The inset shows type 5 'cap+' hairpin element alteration to 'cap–' (yielding type 5$^{cap-}$). (**B**) Gel mobility shift characteristic of complex formation resulting from mixing of type 5 ribozyme with type 1 RNA (equimolar, or with the indicated equivalents). Type 5$^{cap-}$ loses this shift and its susceptibility to type 1 activity enhancement (*Figure 3—figure supplement 1*). Below, shifted band intensities with increasing type 1 addition are plotted (quantified relative to the indicated type 5$^{+1}$ lane intensities, n = 4 ± s.d.), signifying 1:1 heterodimer formation; numerical values are supplied in *Figure 3—source data 1*. (**C**) Type 1 enhancement allows type 5 variants to synthesise long RNAs using triphosphorylated oligonucleotide (11 nt) or short triplet (3 nt) substrates (Sub, 3.6 or 5 μM each, substrate sequences in grey beside lanes; 0.4 μM primer A11/template I-8, 2 μM each Rz, −7°C ice for 16 days). This activity is independent of template tethering (*Wochner et al., 2011*), as comparable synthesis is achieved by versions of type 5 whose 5' regions allow or avoid hybridisation to the template (type 5$^{cis}$ or type 5$^{trans}$ respectively, schematic above, sequences in *Supplementary file 1*). The average extent of ligation at the end of the reaction amongst all junctions in a lane is shown beneath each lane.

DOI: https://doi.org/10.7554/eLife.35255.012

The following source data and figure supplements are available for figure 3:

**Source data 1.** Relative intensities of shifted bands when varying type 1 equivalents.

*Figure 3 continued on next page*

*Figure 3 continued*

DOI: https://doi.org/10.7554/eLife.35255.015

**Figure supplement 1.** Parameters of type 1 activity enhancement.

DOI: https://doi.org/10.7554/eLife.35255.013

**Figure supplement 2.** Type 1 enhancement of parental ribozymes.

DOI: https://doi.org/10.7554/eLife.35255.014

As a first examination of t5$^{+1}$ activity, we revisited triplet-based RNA synthesis on structured templates. To provide a stringent test of template structure inhibition, we now examined hairpin-containing templates (4S, 6S, 8S) with increasing RNA hairpin stability and estimated $T_M$s of up to 93°C (8S). The latter had previously strongly arrested even the most advanced mononucleotide RPRs at higher temperatures (*Horning and Joyce, 2016*). However, using triplets as substrates t5$^{+1}$ robustly copied all of these (*Figure 5a*), even when templates were pre-folded allowing RNA secondary structures to form prior to triplet addition (*Figure 5—figure supplement 1*). The triplet concentration-

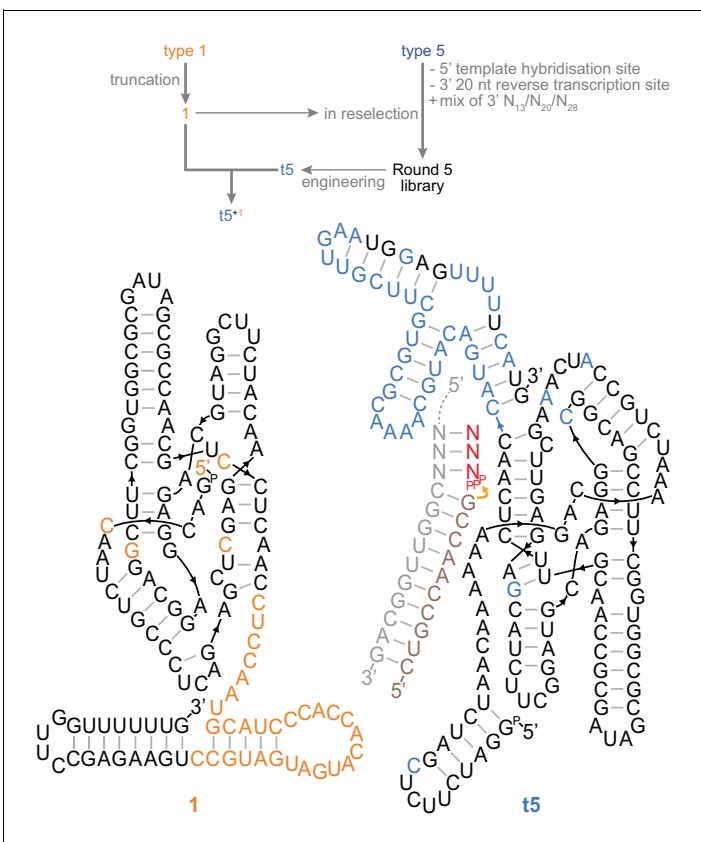

**Figure 4.** A trans-acting heterodimeric triplet polymerase. Top, scheme outlining derivation of the final t5$^{+1}$ triplet polymerase archetype from the type 1 and type 5 RNAs (shown in *Figure 3a*) by reselection using the conditions in *Figure 4—source data 1*. Below, the secondary structure of this ribozyme heterodimer, 135 nt (1) and 153 nt (t5) long, is depicted operating in trans on a non-tethered primer/template duplex. Type 5 3' domain bases that re-emerged after randomisation during reselection are coloured black in the t5 3' domain. Ribozyme development is summarized in *Figure 4—figure supplement 1*; all ribozyme sequences are listed in *Supplementary file 1*.

DOI: https://doi.org/10.7554/eLife.35255.016

The following source data and figure supplement are available for figure 4:

**Source data 1.** Selection conditions of rounds 1–5 of the reselection.

DOI: https://doi.org/10.7554/eLife.35255.017

**Figure supplement 1.** Summary of ribozyme development in this work.

DOI: https://doi.org/10.7554/eLife.35255.018

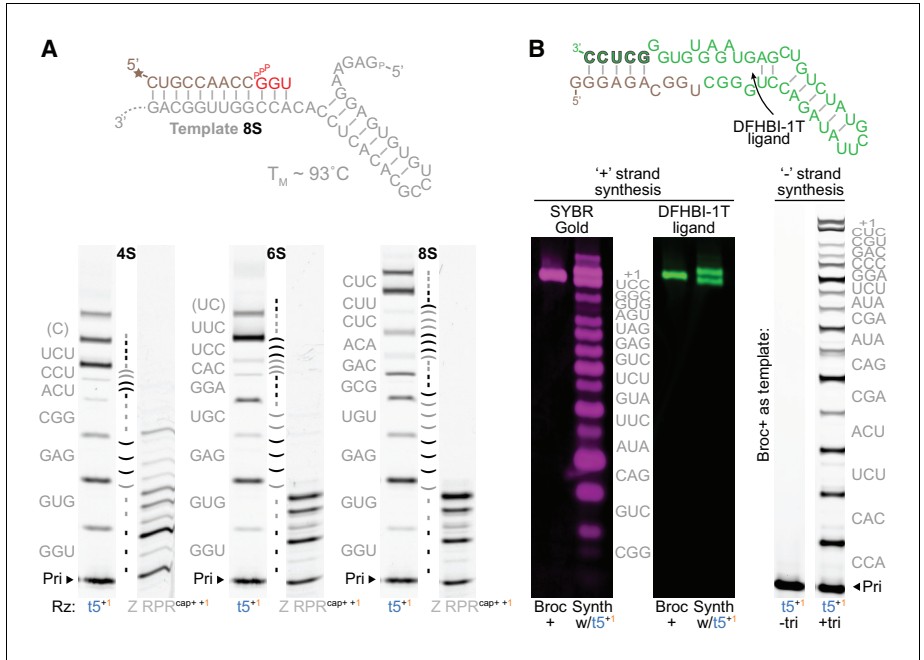

**Figure 5.** Triplet-mediated structured and functional template copying. (**A**) Extension on three structured hairpin templates (4S, 6S, 8S) with increasing stability. Top, the mfold-predicted (*Markham and Zuker, 2005*) structure and $T_M$ of the most stable 8S template; below, primer extensions on these templates by t5$^{+1}$ triplet polymerase (with 5 µM each encoded triplet) or type 1-enhanced Z polymerase ribozyme (Z RPR$^{cap+ +1}$, with 1 mM each NTP) (2 µM ribozyme, 0.5 µM 4S, 6S or 8S template and primer A9, −7°C ice 25 days). The self-complementary region in each template is indicated between each pair of lanes (shaded by triplet), with the encoded triplet substrate sequences at the left (in grey, with 5′ template overhangs in brackets). Syntheses using different substrate compositions and concentrations are shown in *Figure 5—figure supplement 1* and *Figure 5—figure supplement 2*. While all hairpin templates are robustly copied by t5$^{+1}$, synthesis by the Z RPR is completely arrested by the 6S and 8S hairpins. (**B**) Synthesis of the broccoli aptamer. The native secondary structure is shown above (Tan: bases from '+' strand synthesis primer. Green: bases from triplets. Outlined green: primer binding site for the '−' strand synthesis). Below left, t5$^{+1}$-catalysed synthesis of fluorescent broccoli aptamer, run alongside standard (Broc+, synthesized by in vitro transcription), and stained for RNA with SYBR Gold (magenta) or folded with DFHBI-1T ligand (green fluorescence) (2 µM t5$^{+1}$, 1 µM BBrc10/TBrc, 5 µM each triplet (in grey), −7°C ice 22 days). Below right, '−' strand synthesis on Broc + standard (0.5 µM without ligand in ribozyme extension buffer, 0.5 µM FBrcb6 primer, 2 µM t5$^{+1}$, 5 µM each triplet (in grey), −7°C ice 38 days). t5$^{+1}$ is able to synthesise both full-length functional (fluorescent) Broccoli '+' and encoding '−' strands.

DOI: https://doi.org/10.7554/eLife.35255.019

The following figure supplements are available for figure 5:

**Figure supplement 1.** Substrates for structured template copying.
DOI: https://doi.org/10.7554/eLife.35255.020

**Figure supplement 2.** Substrate concentration dependence of structured template copying.
DOI: https://doi.org/10.7554/eLife.35255.021

dependent cooperative structure invasion and unraveling (previously observed with the simpler 0core domain and partly wobble-paired RNA template structures [*Figure 1b,c*]) was recapitulated with t5$^{+1}$ and the highly stable 8S hairpin template (*Figure 5—figure supplement 2*). In contrast, dinucleotide triphosphate substrates yielded extension only up to the structured region (*Figure 5— figure supplement 2*).

We began to explore whether triplet-based RNA synthesis by t5$^{+1}$ might exhibit the generality required not just for synthesis of arbitrary structured sequences, but for replication of functional sequences (requiring synthesis of both '+' and '−' strands). Encouragingly, t5$^{+1}$ could synthesise both a functional fluorescent '+' strand of the 52 nt Broccoli RNA aptamer (*Filonov et al., 2014*) and its encoding '−' strand template from their 13 (+) and 12 (−) different constitutive triplets (*Figure 5b*).

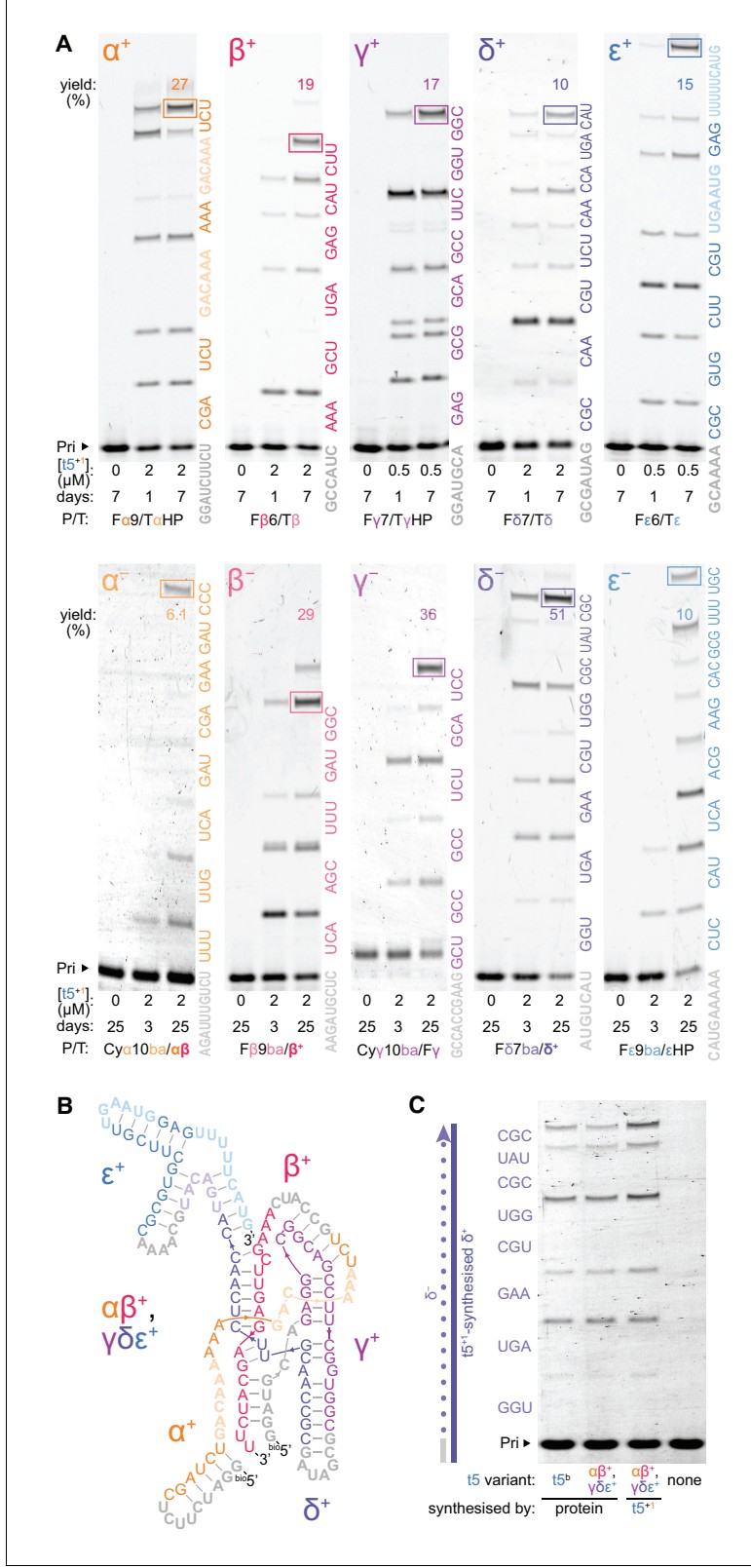

**Figure 6.** Ribozyme self-synthesis and assembly of its own catalytic domain. (**A**) t5$^{+1}$-catalysed syntheses of the five catalytic domain '+' and '−' segments via triplet extension of primers (grey) in −7˚C ice. Triplets are coloured by segment and shown alongside the lanes; longer oligonucleotide substrates (faded) were provided for α$^+$ and ε$^+$ syntheses to combat ribozyme-template pairing as shown in **Figure 6—figure supplement 2** and **Figure 6—**
*Figure 6 continued on next page*

*Figure 6 continued*

*figure supplement 3*. The triplets were supplied at 5 μM ($α^+$ to $ε^+$), 10 μM ($β^-$ to $ε^-$), or 20 μM ($α^-$) each, with 0.5 μM primer/template (P/T; 1 μM for Fβ6/Tβ) and oligonucleotides equimolar to template sites. Use of substrates of more heterogenous compositions and lengths is shown in *Figure 6—figure supplement 4* and *Figure 6—figure supplement 5*, respectively. Densitometry gave yields of full-length products (boxed, by % of total primer), and a geometric mean of the final extents of ligation across all 70 junctions in this self-synthesis context (78%). These segment sequences derive from $t5^b$, a t5 variant with a neutral signature mutation (*Supplementary file 1*). (B) Secondary structure representation of a t5 catalytic domain ($αβ^+/γδε^+$, $t5^b$ sequence), formed via non-covalent assembly of $t5^{+1}$-synthesised '+' strand fragments in *Figure 6—figure supplement 1*, coloured by segment and synthesis substrate as in (A). (C) Activity of ribozyme-synthesised $αβ^+/γδε^+$ (B), compared to protein-synthesised $αβ^+/γδε^+$ and full-length $t5^b$ equivalents. These were assayed for synthesis of a $δ^-$ strand segment on a ribozyme-synthesised $δ^+$ template, with added in vitro transcribed type 1 (2 μM each Rz, 5 μM triplets, 0.5 μM P/T, −7°C 0.25× ice 10 days). The ribozyme-synthesized and assembled $αβ^+/γδε^+$ ribozyme is as active as in vitro transcribed equivalents, and can efficiently utilize ribozyme-synthesized RNA ($δ^+$) as a template.
DOI: https://doi.org/10.7554/eLife.35255.022

The following figure supplements are available for figure 6:

**Figure supplement 1.** Ribozyme catalytic domain self-synthesis and assembly.
DOI: https://doi.org/10.7554/eLife.35255.023

**Figure supplement 2.** Substrate competition attenuates inhibitory $ε^+/ε^-$ pairing during self-synthesis.
DOI: https://doi.org/10.7554/eLife.35255.024

**Figure supplement 3.** Ribozyme stabilisation attenuates inhibitory $δ^+/δ^-$ pairing during self-synthesis.
DOI: https://doi.org/10.7554/eLife.35255.025

**Figure supplement 4.** Ribozyme segment synthesis with random substrate pools.
DOI: https://doi.org/10.7554/eLife.35255.026

**Figure supplement 5.** Ribozyme segment synthesis with mixed length substrate pools.
DOI: https://doi.org/10.7554/eLife.35255.027

## Ribozyme sequence self-synthesis and assembly

We next turned to the critical test of generality: could triplet substrates allow self-synthesis? As $t5^{+1}$ currently lacks the efficiency to synthesise RNAs its own length, we divided the catalytic t5 ribozyme into five segments α, β, γ, δ and ε. This segmentation strategy (akin to that used by some RNA viruses e.g. influenza) could reduce tertiary structures (*Doudna et al., 1991*; *Mutschler et al., 2015*) and ease product separation during RNA replication (*Szostak, 2012*). Starting from ~8 nt RNA primers, $t5^{+1}$ achieved synthesis of the $β^+$, $γ^+$, and $δ^+$ segments from their constitutive triplets as well as all of the '−' strand segments $α^-$, $β^-$, $γ^-$, $δ^-$ and $ε^-$, but required some triplets pre-linked (as e.g. hexa- or nonanucleotides) for synthesis of full-length $α^+$ and $ε^+$ segments (*Figure 6a*).

Operating across 70 distinct ligation junctions in these reactions including AU-rich sequences, $t5^{-+1}$ demonstrates the sequence generality for self-synthesis using triplet substrates. Notably, the average extent of ligation per junction during synthesis of t5 '+' and '−' strands (78%) was similar to that observed when $t5^{+1}$ used an unstructured model template (74%, *Figure 3c*) upon which the parental Z and other RPRs excel (*Attwater et al., 2013b*; *Horning and Joyce, 2016*).

At this point, we tested whether the broad oligonucleotide ligation capacity of $t5^{+1}$ (*Figure 3c*) might allow assembly of synthesised '+' strand segments. Indeed, $t5^{+1}$ could assemble these into $αβ^+$ and $γδε^+$ fragments, guided only by partially overlapping '−' strands (*Figure 6b*, *Figure 6—figure supplement 1*). Through non-covalent association (*Vaish et al., 2003*; *Mutschler et al., 2015*), the ribozyme-synthesised $αβ^+$ and $γδε^+$ fragments spontaneously reconstituted a new catalytically active triplet polymerase ribozyme (with in vitro transcribed type 1 RNA). We found that this synthesis product could regenerate fresh $δ^-$ segment using $t5^{+1}$ ribozyme-synthesised $δ^+$ (left over from ribozyme assembly) as a template (*Figure 6c*), recapitulating elements of a self-replication cycle. However, while the $t5^{+1}$ ribozyme displays a nascent capacity for templated synthesis of its own catalytic domain '+' strands (and '−' strands), efficiency of both segment synthesis and assembly will need to be increased significantly to realise a full self-replication cycle (which would also require synthesis and replication of the type 1 subunit).

## Primer-free RNA synthesis

Templated '+' strand self-synthesis is a central element of ribozyme self-replication. However, a limitation of our above strategy in the context of triplet-based self-replication is the continued requirement for some pre-synthesized longer oligonucleotides to act as primers and occasional substrates (together providing here the equivalent of ~25% of triplet junctions pre-ligated). In particular, some specific oligonucleotide substrates were required for efficient synthesis of α+ and ε+ segments to compete out inhibitory mutual hybridisation between '−' strand template and corresponding '+' strand unstructured elements in the t5 ribozyme (*Figure 6—figure supplement 2*). In vitro selections that stabilise the ribozyme tertiary structure (*Figure 6—figure supplement 3*) may contribute to attenuating this requirement. Additionally, more concentrated triplet substrates can successfully compete with ribozyme unstructured elements for hybridization to '−' strand templates (*Figure 6—figure supplement 2*).

The majority of specific oligonucleotides, however, were provided as primers to initiate syntheses, as required by all RPRs akin to the activity of replicative polymerases in biology. As a consequence of this, the priming sequence would effectively be excluded from evolution during replication. Furthermore, RNA oligonucleotides able to act as specific primers are unlikely to be prevalent in prebiotic substrate pools, and their depletion during successive replication cycles could lead to loss of sequence at genome ends. This 'primer problem' has previously been noted in the context of nonenzymatic replication (*Szostak, 2012*) as one of the fundamental obstacles to RNA self-replication.

Unexpectedly, triplet substrates provide a route to bypass the 'primer problem'. We observed that t5$^{+1}$ can extend primers bidirectionally, in both the canonical 5′−3′ as well as the reverse 3′−5′ directions (*Figure 7a*). This allows not only completion of RNA synthesis from either template end but also initiation from anywhere along a template, potentially allowing non-classical hierarchical or distributive RNA replication schemes as previously proposed (*Szostak, 2011*). Given this flexibility, we wondered if t5$^{+1}$ even had a requirement for a primer oligonucleotide. Indeed, this triplet polymerase could achieve 'primer free' RNA synthesis (whereby synthesis is presumably initiated by ligation of adjacent triplets anywhere on the template), as exemplified here for the β$^{+}$ segment (*Figure 7b*), as well as 'primer free' RNA replication as shown for the '+' and '−' strands of the γ segment, which can be replicated using triplets alone (*Figure 7c*).

Thus, the capacity of triplet substrates to pre-organise themselves on a template not only enables replication of structured templates but also allows complete copying of some RNA sequences exclusively from triplet building blocks, suggesting an alternative to the canonical end-primed replication strategies inspired by PCR. Such a ribozyme operating in a more distributive polymerisation mode might be able to replicate RNA sequences directly from the putative pools of short random RNA oligonucleotides furnished by prebiotic chemistry.

## Fidelity of triplet-based RNA synthesis

Next, we investigated the consequences of using analogues of such prebiotic pools as a source of substrates for the t5$^{+1}$ triplet polymerase ribozyme. Random sequence triplet pools ('$^{PPP}$NNN', comprising equimolar amounts of all 64 triplets) could be used as substrates by t5$^{+1}$ in segment syntheses in place of defined triplet sets (*Figure 6—figure supplement 4*). Furthermore, extension activity remained robust upon pool supplementation with noncanonical dinucleotide and mononucleotide substrates (*Figure 6—figure supplement 5*).

However, a replicase must incorporate the correct template-complementary substrate from random sequence pools, or genetic information may become irretrievably corrupted during replication (*Eigen, 1971*). Sequence fidelity is therefore a critical parameter of RNA replication. The fidelity challenge is exacerbated in triplet-based RNA replication by the need to discriminate between 64 distinct substrates; indeed, a previous investigation into the incorporation of individual trinucleotides indicated that misincorporations could outstrip cognate incorporation for some triplets (*Doudna et al., 1993*).

In order to assess the fidelity of triplet polymerase ribozymes of widely differing activity, we identified the triplets incorporated from random $^{PPP}$NNN triplet pools using 12 different compositionally representative N′N′N′ triplet sequences as templates. These were examined in a consistent sequence context (5′-GGG-N′N′N′-GGG-3′) and collated, which allowed an estimation of ribozyme

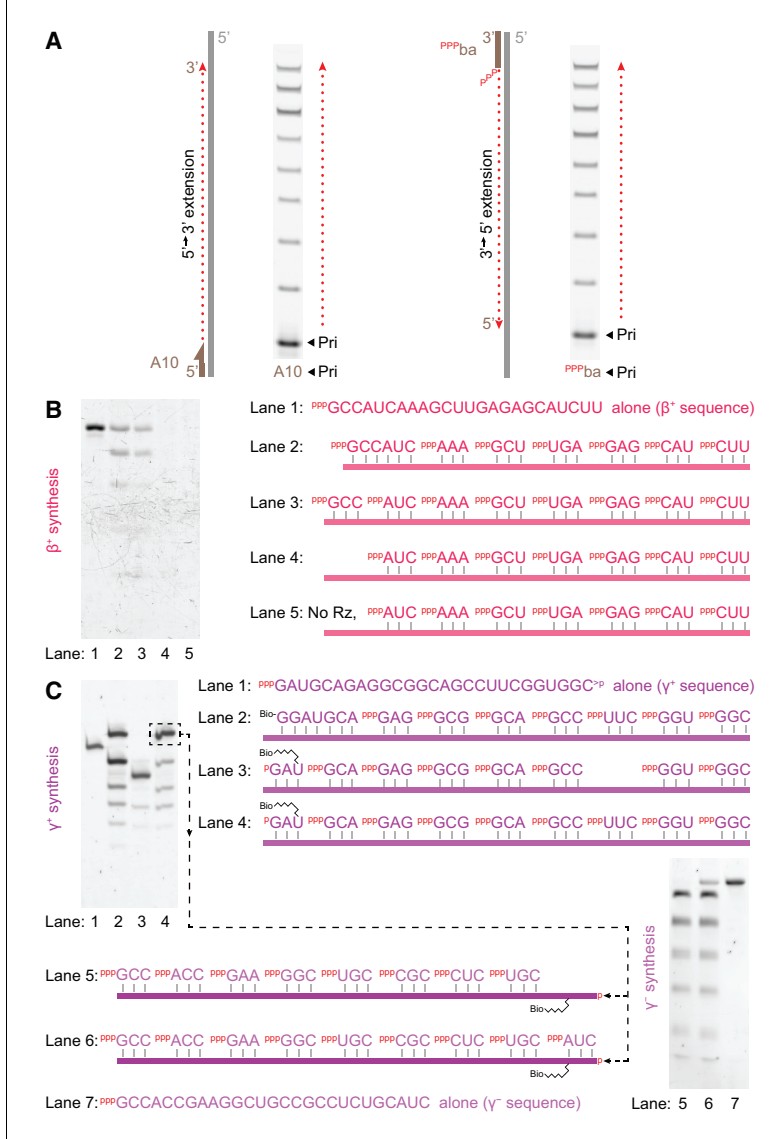

**Figure 7.** Triplet-initiated template sequence copying. (**A**) Extension by t5$^{+1}$ of fluorescein-labelled primers bound to either the 3′ (A10) or 5′ ($^{PPP}$ba) ends of a template (T8GAA, 10 μM $^{PPP}$GAA, 0.5 μM/RNA, −7°C ice 69 hr), demonstrating extension in either 5′−3′ (A10) or 3′−5′ ($^{PPP}$ba) directions. (**B**) Synthesis of β$^{+}$ on Tβ template via t5$^{+1}$-catalysed polymerisation of the substrates indicated on the right (2 μM t5$^{+1}$, 5 μM each triplet, 0.5 μM template (lane 2: 0.5 μM hexanucleotide), −7°C ice 9 days). Extension products in lanes 2–5 were eluted from template, PAGE-separated and SYBR-Gold stained alongside in vitro transcribed full-length segment control (lane 1). Lane 3 shows full-length synthesis of β$^{+}$ segment from triplets alone. (**C**) Triplet-based replication of the γ segment. Top left, synthesis of γ$^{+}$ from the indicated substrates (1 μM t5$^{+1}$, 5 μM each triplet, 2 μM TγHP template (lane 2: 2 μM Bioγ7 primer), −7°C ice 7 days). Biotinylated extension products in lanes 2–4 were isolated from template, PAGE-separated and SYBR-Gold stained alongside in vitro transcribed staining marker (Mγ$^{+}$m1, lane 1). This indicated 10% yield (per template) of full-length synthesis of γ$^{+}$ segment from triplets alone (the final band in lane 4), which was purified for use as a template in γ$^{-}$ segment synthesis (bottom right, 1 μM t5$^{+1}$, 5 μM each triplet, 0.05 μM template, with 0.05% Tween-20, −7°C ice 27 days). Extension products in lanes 5 and 6 were eluted from template, PAGE-separated and SYBR-Gold stained alongside in vitro transcribed full-length segment control (Mγ$^{-}$m1, lane 7). This indicated 6% yield (per template) of full-length synthesis of γ$^{-}$ segment from triplets alone (the final band in lane 6).

DOI: https://doi.org/10.7554/eLife.35255.028

misincorporation tendencies. On average, the starting Zcore ribozyme exhibited ~91% fidelity per position (*Figure 8a*), lower than that described for RPRs (92% – 97% [*Attwater et al., 2013b; Horning and Joyce, 2016*]). Furthermore, its accuracy exhibited a pronounced downward gradient from the first (5') to the third (3') triplet position, highlighting escalating risks to fidelity of synthesis founded on longer building blocks.

To investigate if ribozymes could exhibit higher triplet incorporation fidelity, we had included a persistent adaptive pressure for fidelity during in vitro evolution, spiking in an excess of mispairing 3'-deoxy 'terminator' triplets from round nine onwards, precluding recovery of ribozymes that incorporated these mispairs (*Figure 2—source data 1*, *Figure 4—source data 1*, *Figure 2—figure supplement 2*). This yielded reshaped and improved fidelity profiles in the 'surviving' type 2–6 ribozymes (*Figure 8a*). Notably, the final t5$^{+1}$ ribozyme achieves an average positional fidelity of 97.4% using $^{PPP}$NNN in this sequence context, higher than the best RPR fidelity with NTPs under comparable eutectic conditions (*Attwater et al., 2013b*). Deep sequencing of internal triplet positions of a defined sequence (β$^+$ segment) synthesised by t5$^{+1}$ using $^{PPP}$NNN indicated similar aggregate fidelity could be achieved during longer product synthesis excluding the final triplet (*Table 1*).

## Molecular basis of triplet polymerase ribozyme fidelity

Having established that accurate triplet-based copying is possible (in at least some sequence contexts), we sought to understand how the triplet polymerase ribozyme achieves it. Investigating the fidelity contributions of different t5$^{+1}$ ribozyme components, we found that the type 1 RNA cofactor did not contribute; rather, fidelity gains appeared to be mediated by the newly-evolved t5 'ε' 3'-domain, as its deletion (yielding the truncated 'αβγδ' ribozyme) reverted the fidelity profile towards that of Zcore (*Figure 8—figure supplement 1*). Presence of the ε domain did not uniformly increase fidelity, but selectively reduced the most acute errors at the second and third triplet positions (with over 10-fold reductions for some errors, *Figure 8b*, *Figure 8—figure supplement 2*). Overall error rates at the second and third triplet positions were reduced by 4-fold and 9-fold compared to Zcore (*Figure 8a*), though increased (1.3-fold) at the first triplet position due to a localised asymmetric tolerance of G:U wobble pairing (*Figure 8b*). The ε domain fidelity function is contingent upon the presence of a downstream triplet, operating only with basal fidelity for final triplet incorporation (*Figure 8—figure supplement 3*).

Dissecting the molecular determinants of the fidelity phenotype, we found that using triplet substrates modified at the third position with a 2-thiouracil in place of a uracil (disrupting minor groove hydrogen bonding capabilities) rendered the ε fidelity domain unable to discriminate mismatches (*Figure 8c*, *Figure 8—figure supplement 3*). Previously, a similar replacement of a uracil 2-keto group with a 2-thio modification had been shown to impair Z RPR activity when present upstream in the primer/template region (*Attwater et al., 2013a*), where Z is thought to rely upon sequence-general minor groove contacts through an 'A-minor' motif (*Shechner et al., 2009*). Modification at the third triplet position reverts ε's divergent effects on fidelity at the adjacent second and the distal first triplet positions (*Figure 8—figure supplement 3*); disruption of this minor groove contact site thus abolishes overall ε fidelity domain operation. ε sensitivity to minor groove composition may be critical to its recognition of cognate Watson-Crick base pairs, reminiscent of *Tetrahymena* group I intron folding (*Battle and Doudna, 2002*) and the decoding centre of the ribosome (which also tolerates wobble pairing at the analogous (5') triplet position) (*Ogle et al., 2001*).

## Systems-level properties of triplet pools

An important contribution to triplet fidelity also appears to arise from unexpected behaviours of the triplet substrates themselves. We observed that in some direct pair-mispair triplet contests, inclusion of their complementary triplets caused a striking (~3 fold) drop in misincorporation errors (*Figure 9*). A potential explanation may arise from differential formation of triplet:anti-triplet dimers in the reaction: for example, more extensive $^{PPP}$GCC:$^{PPP}$GGC (than $^{PPP}$ACC:$^{PPP}$GGU) dimer formation would selectively reduce the effective concentration of free $^{PPP}$GCC vs. $^{PPP}$ACC upon inclusion of their complementary $^{PPP}$GGC and $^{PPP}$GGU.

These pairwise reductions were recapitulated in the presence of random $^{PPP}$NNN substrate pools (*Figure 9*). Indeed, counterintuitively, raising $^{PPP}$NNN concentrations from 0.5 to 5 μM each almost

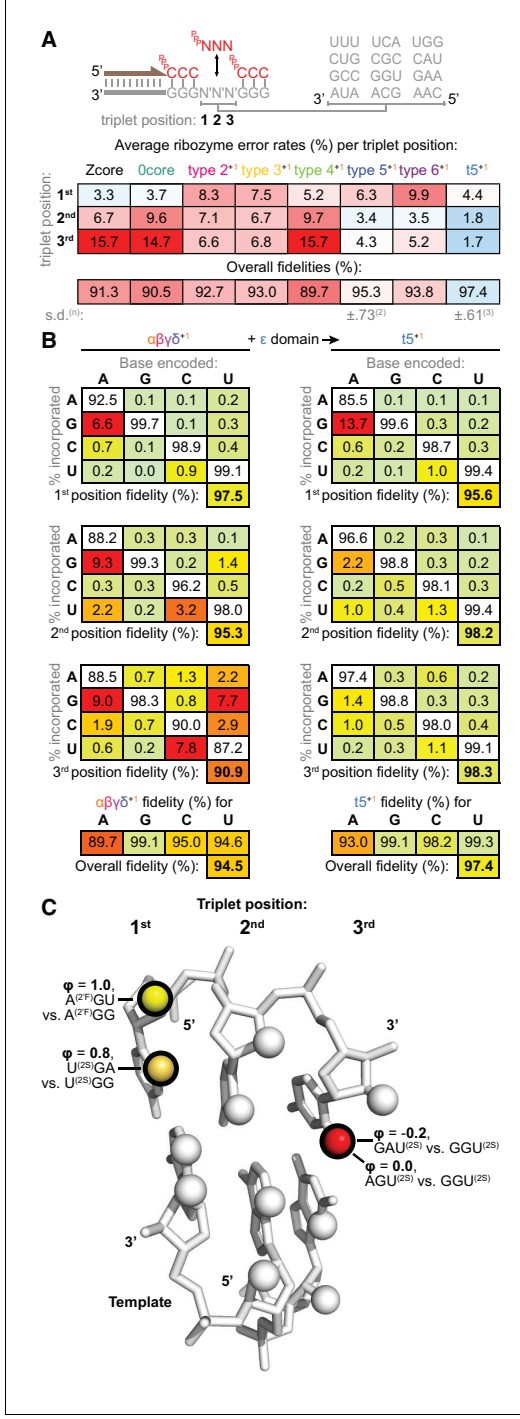

**Figure 8.** Fidelity of ribozyme-catalysed triplet polymerisation. (**A**) To estimate its fidelity, each ribozyme was provided with an equimolar mix of all 64 triplet substrates ($^{PPP}$NNN at 5 μM each) for primer extension using templates containing twelve representative trinucleotide sequences (N'N'N'). Deep sequencing of extension products identified the triplets added opposite each template trinucleotide, yielding position-specific error tendencies; the overall fidelity was calculated as a geometric mean of positional

*Figure 8 continued on next page*

halved the overall error rate (*Figure 9—figure supplement 1*). Although diverse effects upon individual misincorporations were observed, this fidelity enhancement was driven by pronounced reductions in errors where the mismatched triplet has a high GC content compared to the cognate triplet, including common G-U wobble mispairs (*Figure 9—figure supplement 1*). Dimer formation among $^{PPP}$NNN substrate pools would be expected to selectively buffer the free concentrations of the more strongly-pairing GC-rich triplets, which could promote both fidelity and sequence generality through normalization of triplet availability against template (and complementary triplet) binding strength.

Indeed, more efficient, higher fidelity segment synthesis was observed when partially mimicking this outcome using an $^{PPP}$NNN pool formulated with a reduced G content (*Figure 6—figure supplement 4*, *Table 1*). In a prebiotic scenario, substrate pool composition would have been determined by the abundance and nontemplated polymerization tendencies of the different nucleotides; large biases in these could skew triplet compositions or deplete a triplet (resulting in mismatch incorporation). However, the potential for replication to proceed in different triplet registers may provide a degree of resilience towards such biases.

## Discussion

Here, we describe the discovery and characterization of a ribozyme (t5$^{+1}$) with a robust ability to polymerize RNA trinucleotide triphosphate (triplet) substrates. Unusually, this triplet polymerase ribozyme comprises a heterodimer of a catalytic triplet polymerase subunit (t5) and a non-catalytic RNA cofactor (type 1), which enhances triplet polymerase activity and abrogates the need for template tethering. Such a quaternary structure - involving a heterodimer of a full-length and a truncated subunit - is reminiscent of the processivity factors of some proteinaceous polymerases such as the heterodimeric p66/p51 HIV reverse transcriptase holoenzyme (*Huang et al., 1992*). There are multiple examples of dimerization in RNA evolution - such as the VS ribozyme (*Suslov et al., 2015*), retroviral RNA genome dimerization (*Paillart et al., 2004*), in vitro evolved heterodimeric RNA liposome binders (*Vlassov et al., 2001*), and recently the homodimeric CORN fluorescent RNA aptamer (*Warner et al., 2017*). However, the spontaneous emergence of a general, mutualistic RNA cofactor has not previously been observed for

*Figure 8 continued*

errors at each triplet position (n and s.d. of this value shown for ribozymes assayed multiple times, see *Figure 8—source data 1* for analysis of collated errors). The triplet polymerases exhibit diverse fidelity profiles; fidelity profiles of other type 5 variants are shown in *Figure 8—figure supplement 1*. (B) Collation of error rates by base type and position for type 5 with (t5$^{+1}$) and without ($\alpha\beta\gamma\delta^{+1}$) the ε fidelity domain. Positional and overall fidelities are calculated as geometric means (see *Figure 8—source data 1*); individual positional fidelities are plotted in *Figure 8—figure supplement 2*. (C) Schematic summary of effects of triplet minor groove modification upon the fidelity phenotype. In the depicted trinucleotide RNA duplex segment, spheres represent minor groove groups potentially available for hydrogen bonding in a sequence-general manner. For three of these groups (highlighted in black), we assayed whether their modification in substrates (2'F = 2' fluoro, 2S = 2-thio) affected the fidelity domain's mismatch discrimination capabilities (detailed in *Figure 8—figure supplement 3*, with data and calculations in *Figure 8—source data 2*). These groups are labelled with the fraction of fidelity phenotype retained (ϕ) when discriminating between the indicated modified substrates. Colour reflects the impact of that group's modification upon the fidelity phenotype, with red denoting a strong disruptive effect, and yellow weak or negligible effects.
DOI: https://doi.org/10.7554/eLife.35255.029

The following source data and figure supplements are available for figure 8:

**Source data 1.** Analysis of collated errors by ribozymes in the fidelity assay.
DOI: https://doi.org/10.7554/eLife.35255.033
**Source data 2.** Calculation of residual fidelity phenotypes in *Figure 8c*.
DOI: https://doi.org/10.7554/eLife.35255.034
**Figure supplement 1.** Fidelity of type 5 variants.
DOI: https://doi.org/10.7554/eLife.35255.030
**Figure supplement 2.** Fidelity domain influence upon template- and position-specific error rates.
DOI: https://doi.org/10.7554/eLife.35255.031
**Figure supplement 3.** Determination of residual fidelity phenotype when using minor groove-modified substrates.
DOI: https://doi.org/10.7554/eLife.35255.032

ribozymes and may suggest an underappreciated dimension to the evolutionary dynamics of ribozyme pools under stringent adaptive pressures. Indeed, the extinction of previously dominant species in the selection that were unable to benefit from type 1 enhancement (e.g. type 0, see *Figure 3—figure supplement 2*) and succession with cooperative RNA species (*Vaidya et al., 2012*) illustrates the potential for such symbioses to shape RNA molecular ecologies.

The t5$^{+1}$ ribozyme's principal current shortcoming is its low catalytic efficiency. In the optimal context for mononucleotide polymerase ribozymes, this triplet polymerase heterodimer yields ~4 fold more unligated junctions than the RPR tC19Z (*Attwater et al., 2013b*), which itself is 240-fold slower than the currently most advanced RPR 24–3 (*Horning and Joyce, 2016*). Yet despite this modest catalytic power, t5$^{+1}$ displays much enhanced generality in RNA synthesis and now achieves both copying of previously intractable structured RNA templates, and templated synthesis and assembly of an active '+' strand copy of its catalytic domain, suggesting key contributions of the triplet substrates themselves.

Indeed, one of the main findings of our work are the compelling advantages that triplet substrates appear to offer for sequence general RNA replication. For instance, when binding templates, triplets incur a lower entropic cost per position compared to canonical mononucleotides (thus aiding copying of sequences rich in weakly pairing A and U bases), with particularly helpful stability contributions from intra-triplet base stacking (*Eigen, 1971*). Furthermore, energetically favourable inter-triplet stacking interactions appear to instigate cooperative binding and unfolding of even highly stable RNA template structures (*Figures 1b* and *5a*) upon reaching the required substrate concentration threshold. In our work, this process is aided by the cold temperature and solute concentration effects of eutectic ice phase formation (*Attwater et al., 2010*; *Mutschler et al., 2015*). Counterintuitively, a general solution to the copying of structured RNAs arises not from conditions that disfavour base-pairing (which would also hinder substrate binding), but rather from conditions that promote it.

Together these favourable molecular traits serve to pre-organize the template towards a double-stranded RNA duplex with triplet junctions poised for ligation. A triplet/template duplex presents a more ordered, regular target for sequence-general ribozyme docking (by e.g. the ε domain) than a single stranded template (variably prone to secondary structure formation or sequence-specific interactions with the ribozyme [*Wochner et al., 2011*]). Such general duplex interactions also underlie other notable features observed in our triplet-based RNA synthesis such as in trans template binding

**Table 1.** Sequencing of ribozyme-synthesised β+ segment.

Shown are the individual base fidelities (%) along the β+ sequences (top) synthesised by t5+1, using the six specific triplets (tri), or random (PPPNNN) or compositionally-biased random (low-G PPPNNN, see *Figure 6—figure supplement 4*) substrate pools, from Fβ6 primer (the first six positions at the left) with template Tβ (1 μM each RNA, 13 days −7°C ice). For their sequencing, extension products were eluted from templates, and full-length products were gel-purified, ligated to adaptor, reverse-transcribed and PCR amplified. For compositional analysis, a small percentage of unrelated amplified products were excluded (those with >9 mutations vs. the expected β+ sequence; similar levels were excluded if a > 6 mutation threshold was applied, 0.2%/0.2–3.7%/4.2% and 3.7%/3.8% for tri & PPPNNN and low-G PPPNNN). These sequences mostly appeared to derive from off-target priming and extension of Fβ6 on the ribozyme in the presence of PPPNNN. The sequencing of products generated from specific triplets provides an estimate of background error arising from amplification and sequencing. The final triplet constitutes an error hot-spot - likely to mutate to a more mutationally stable triplet during self-replication - exacerbated in PPPNNN samples by the inability of the fidelity domain to operate in the absence of a downstream triplet (*Figure 8—figure supplement 3*). The geometric average of internal triplet position fidelities is used to gauge overall t5+1 fidelity during RNA synthesis. While overall fidelity drops from defined to random triplets (98.8 to 96.7%), much of this loss in fidelity can be recovered by adjusting the triplet composition to a low-G random pool, where reductions in G-U wobble pairing more than compensate for increases in rarer misincorporations opposite template C.

| | G | C | C | A | U | C | A | A | A | G | C | U | U | G | A | G | A | G | C | A | U | C | U | U | Internal triplets' average: |
|---|---|---|---|---|---|---|---|---|---|---|---|---|---|---|---|---|---|---|---|---|---|---|---|---|---|
| 10 μM each tri: | | | | | | | 93.3 | 99.4 | 99.5 | 97.8 | 99.8 | 99.4 | 99.6 | 97.9 | 99.1 | 99.3 | 99.5 | 99.1 | 99.8 | 98.9 | 99.5 | 98.2 | 96.0 | 96.3 | 98.79 |
| 10 μM each PPPNNN: | | | | | | | 92.8 | 97.0 | 98.8 | 99.2 | 99.5 | 99.3 | 99.1 | 97.8 | 98.6 | 99.4 | 99.1 | 98.7 | 94.3 | 81.4 | 96.6 | 97.4 | 59.8 | 42.3 | 96.65 |
| 10 μM average, low-G PPPNNN: | | | | | | | 97.3 | 98.2 | 99.5 | 97.3 | 99.7 | 99.4 | 99.6 | 97.8 | 99.0 | 99.3 | 98.9 | 98.8 | 97.6 | 97.3 | 98.8 | 97.4 | 73.2 | 43.2 | 98.56 |

DOI: https://doi.org/10.7554/eLife.35255.035

(*Figure 3c*) as well as the capacity for bidirectional (5′−3′/3′−5′) and primer-free RNA synthesis (*Figure 7*).

Contrary to expectations RNA-catalyzed triplet polymerisation can proceed with a fidelity matching or exceeding even the best mononucleotide RNA polymerase ribozymes (*Attwater et al., 2013b*; *Horning and Joyce, 2016*). t5+1 ribozyme fidelity is due to both a readout of cognate minor groove interactions by the ribozyme ε domain (*Figure 8*) and an unanticipated fidelity boost arising from systems-level properties of triplet pools, that appear to normalize the availability of free triplet (and potentially longer oligonucleotide) substrates against their base-pairing strength (*Figure 9*). Though further work will be required to characterize triplet pool properties, they likely involve formation of cognate or near-cognate triplet:anti-triplet interaction networks, as formation of tRNA dimers via cognate anticodon:anticodon interactions has been observed in a similar concentration range (*Eisinger and Gross, 1975*).

While phylogenetically unrelated, mechanistic analogies between the triplet polymerase ribozyme and the ribosome are apparent. Both are RNA heterodimers that operate in a triplet register along a single-stranded RNA template, whilst enforcing a minor-groove mediated pattern of triplet or anticodon readout (including tolerance of 5′ wobble pairing), suggestive of convergent adaptive solutions to the challenges of replication and decoding. It has long been speculated that the decoding centre of the small ribosomal subunit might have had its origins in an ancestral RNA replicase, but the implied triplet-based character of such a replicase was conspicuously discordant with modern mononucleotide-based replication (*Weiss and Cherry, 1993*; *Poole et al., 1998*; *Noller, 2012*). The utility of triplets as substrates for RNA synthesis and self-synthesis described herein suggests that these early ideas deserve to be reconsidered. In the context of initial uncorrelated evolution of the small and large ribosomal subunits (*Petrov et al., 2015*), it is tempting to speculate that an early reliance upon triplets in RNA replication could have inadvertently supplied a decoding center for translation.

In conclusion, the unexpected emergent properties of triplets – including cooperative binding and unfolding of structured RNA templates, enhanced incorporation of AU-rich substrates, and error attenuation (resulting from triplet pool interaction networks) – argue that short RNA oligonucleotides may represent predisposed substrates for RNA-catalyzed RNA replication. Some of these benefits might also extend to codon/anticodon dynamics in early translation, and to the non-enzymatic

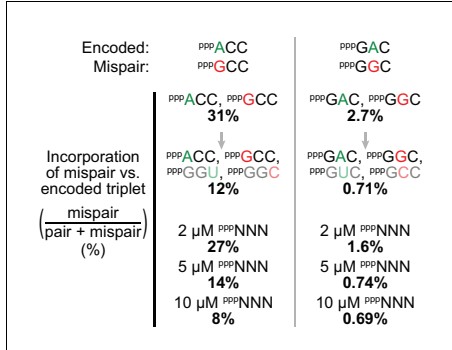

**Figure 9.** Substrate pool interactions improve triplet fidelity. Applying the fidelity assay (**Figure 8a**, using t5$^{+1}$) to single templates with only an encoded triplet and a mispairing one as substrates (at 5 µM each), we observed that relative mispair incorporation was proportionally reduced (by 61% (left) and 73% (right)) upon introduction of complementary triplets. Using all 64 triplets ($^{PPP}$NNN) has an analogous effect upon these pair/mispair comparisons with fidelity progressively improved upon increasing overall $^{PPP}$NNN concentrations, with examples of effects on other triplets and overall fidelity presented in **Figure 9—figure supplement 1**, and comprehensive error rates and ratios in **Figure 9—source data 1**.
DOI: https://doi.org/10.7554/eLife.35255.036
The following source data and figure supplement are available for figure 9:

**Source data 1.** Collated error rates and ratios at different substrate concentrations.
DOI: https://doi.org/10.7554/eLife.35255.038
**Figure supplement 1.** Influence of random triplet pool concentrations upon triplet misincorporation tendencies.
DOI: https://doi.org/10.7554/eLife.35255.037

replication of RNA (**Szostak, 2012**), where downstream trinucleotides have recently been shown to enhance incorporation of preceding activated mononucleotides both through stacking and positioning effects (**Vogel et al., 2005**; **Zhang et al., 2018**) and the formation of a highly reactive intermediate (**Prywes et al., 2016**; **O'Flaherty et al., 2018**). Taken together, the interaction of triplet substrate pools with RNA templates promotes uncoupling of an RNA's sequence (i.e. information content, and associated folding tendencies) from its replicability, thereby enhancing RNA's capacity to serve as an informational polymer.

## Materials and methods

### Templated RNA-catalysed RNA synthesis

Standard ribozyme activity assays (modified where specified) comprise 5 pmol of each ribozyme annealed in 2.5 µl water (80°C 2 min, 17°C 10 min), with 2 µl of 1 M MgCl$_2$ and 0.5 µl of 1 M tris•HCl pH 8.3 (at 25°C, pH raised to 9.2 at −7°C) then added on ice, and left for >5 min to ensure folding. This was added to 5 pmol each of primer and template and 50 pmol of each triplet pre-annealed in 5 µl water, then frozen on dry ice (10 min) and incubated at −7°C in a R4 series TC120 refrigerated cooling bath (Grant (Shepreth, UK)) to allow eutectic phase formation and reaction.

Final pre-freezing concentrations of components are displayed throughout (in this example, yielding 0.5 µM ribozyme/primer/template, 5 µM each triplet, 200 mM MgCl$_2$, 50 mM tris•HCl pH 8.3). Supercooled reactions (**Figure 1c**) remained liquid by omitting the dry-ice freezing step, maintaining these concentrations. Ice crystal formation upon eutectic phase equilibration, however, concentrates all solutes ~4–5 fold (**Attwater et al., 2010**) to their final operational levels and cooling elevates tris-buffered pH to ~9.2.

Some substrate mixes (e.g. $^{PPP}$NNN) led to a higher final reaction volume, but eutectic phase equilibration restored standard operational concentrations, also applicable to the four-fold-diluted extensions with the fragmented ribozyme (**Figure 6c**). These used 2 pmol each ribozyme/fragment annealed in 3.25 µl 62 mM MgCl$_2$, 15 mM tris•HCl pH 8.3 (37°C 5 min, ramped to 4°C at 0.1°C/s, 4°C 10 min), with pre-annealed primer/template/substrates (0.5/0.5/5 pmol) added in 0.75 µl water. These reactions, and preparative syntheses (**Figure 6—figure supplement 1**, **Figure 7c**), were supercooled at −7°C followed by ice crystal addition for quick freezing and optimal activity.

**Figure 1c** extensions were set up by adding buffer, then RNAs (preannealed together, 0.1 µM final concentrations) to triplets. RNAs for ε$^+$ syntheses were chilled on ice instead of annealing, with ribozyme/MgCl$_2$/tris•HCl pH 8.3 mixed with the other RNAs at −7°C. Oligonucleotide substrates were added equimolar to template binding sites in the primer/template/substrate anneal. NTPs, on the other hand, were added with the MgCl$_2$/tris•HCl pH 8.3 to the ribozyme polymerase.

## Extension product separation

At the end of standard incubations, reactions were thawed and 2 µl aliquots added to stop buffer (1 µl 0.44 M EDTA (pH 7.4), with urea to a 6 M final concentration and a 10–20 fold molar excess over template of complementary competing oligonucleotide (see *Supplementary file 3*) to prevent long product/template reannealing). Samples were denatured (94°C 5 min) and RNAs separated by 8 M urea 1 × TBE denaturing PAGE.

To avoid using potentially confounding competing oligonucleotide when purifying extension products, reactions with a biotinylated primer or template (stopped as above) could be purified by bead capture using MyOne C1 (Invitrogen) streptavidin-coated paramagnetic microbeads (using 5 µg pre-washed beads per pmol biotinylated RNA) in 0.5 × − 0.8 × bead buffer (BB: 200 mM NaCl, 10 mM tris•HCl pH 7.4 (at 25°C), 1 mM EDTA, 0.1% Tween-20). After washing twice in BB to remove unbound components, beads were incubated (1 min) in 25 mM NaOH, 1 mM EDTA, 0.05% Tween-20 to denature the duplexes (*Horning and Joyce, 2016*). To recover biotinylated extension products (e.g. *Figure 5b* left panels, *Figure 6—figure supplement 1* $\alpha\beta^+/\gamma\delta\epsilon^+$, *Figure 7c* left panel) the supernatant was discarded, and beads were washed first in BB with 200 mM tris•HCl pH 7.4, then in BB, then heated (94°C 4 min) in 95% formamide, 10 mM EDTA to release primers for urea-PAGE. To recover extension products bound to biotinylated templates (e.g. *Figure 6—figure supplement 1* $\beta^+$, $\delta^+$, $\epsilon^+$, *Figure 7c* right panel) the supernatant was removed, neutralized with 500 mM tris•HCl pH 7.4, spin-concentrated using Ultracel 3K filters (MerckMillipore, UK), recovered and denatured in 6M urea/10 mM EDTA before urea-PAGE. $\beta^+$ synthesised in *Figure 6—figure supplement 1* was not spin-concentrated, leading to a lower recovery yield; $\delta\epsilon^+$ synthesis was denatured directly from the ligation reaction in 60% formamide with excess EDTA.

For gel mobility shift assays (*Figure 3b*), ribozymes were mixed at 0.5 µM, pre-annealed and buffer added on ice as for extension reactions, then mixed with 5 × loading buffer (50% glycerol, 250 mM tris•HCl pH 8.3, 125 mM MgCl$_2$) for separation by native PAGE (0.5 × TB, 8% 59:1 acrylamide:bisacrylamide, 25 mM MgCl$_2$, run in a Hoefer SE600 Chroma (ThermoFisher, Waltham, USA) (upper chamber: 0.5 × TB 50 mM NaOAc, lower chamber: 0.5 × TB 25 mM Mg(OAc)$_2$) kept at 4°C in a circulator bath for 6–8 hr at 10 W), then SYBR Gold stained as below.

## Extension product detection, quantification and purification

Fluorescent primer extension products were detected using the appropriate laser wavelength on a Typhoon Trio scanner (GE Healthcare (GE) (Chicago, USA)); gel densitometry allowed quantification of RNA synthesis efficiency. Gel contrasts in figures were linearly adjusted to optimize display of bands of differing intensities.

The gel in *Figure 5b* (middle panel) was washed thrice (5 min) in water, incubated with 10 µM DFHBI-1T ligand in buffer for 20 min to fold full-length broccoli aptamer (as in [*Filonov et al., 2015*]) and scanned. The ligand was then eluted in three 1 × TBE washes (leaving negligible background fluoresence), and stained in 1 × TBE with SYBR Gold (1:10000), washed again, and re-scanned to detect all RNA products (left panel); scans were aligned via an adjacent Cy5-labelled primer extension lane (not shown).

Full-length product yields in the *Figure 6—figure supplement 1* plus-strand syntheses were calculated by running samples of bead-eluted products (or raw reaction for $\delta\epsilon^+$) alongside known amounts of the positive controls indicated, followed by SYBR-Gold staining. To purify, bead-eluted products were run similarly, and excised using UV shadowing. Products were then eluted from the gel fragments in 10 mM tris•HCl pH 7.4, and Spin-X column filtrate (Costar (Sigma-Aldrich, UK)) precipitated in 75% ethanol with 1 µl 1% glycogen carrier (omitted for $\beta^+$). Recovered full-length product yields were calculated similarly to reaction yields for $\alpha\beta^+/\delta\epsilon^+/\gamma\delta\epsilon^+$, or using A$_{260}$s for $\beta^+$, $\delta^+$, $\epsilon^+$.

## Fragment sequencing

Purified ribozyme- and TGK-synthesized $\alpha\beta^+/\gamma\delta\epsilon^+$ fragments were sequenced by first ligating a 3' adaptor (10 U/µl T4 RNA Ligase 2 truncated KQ in 1 × RNA ligase buffer (New England Biolabs (NEB), (Ipswich, USA)) with 15% PEG-8000 and 2 µM AdeHDVLig at 10°C overnight. These reactions were bound to MyOne C1 microbeads (ThermoFisher (Invitrogen)), washed with BB to remove unligated adaptor, and reverse transcribed (50°C 30 min) with 1 µM HDVrec primer using Superscript III (Invitrogen). Beads were washed again then PCR amplified (five cycles with a 40°C annealing step,

then 20 cycles with a 50°C annealing step) using GoTaq HotStart master mix (Promega (Madison, USA)) and 0.8 µM each of primers P3HDV, and P5Xα8 or P5Xγ7, for high-throughput sequencing (Illumina (San Diego, USA) MiSeq or HiSeq) after PCR product agarose gel purification. β$^+$ syntheses' cDNAs were amplified with P3HDV and P5Xβ6.

## Fidelity assay

To estimate RNA synthesis fidelity, ribozymes extended primers using $^{PPP}$NNN on templates encoding CCC-XXX-CCC, where XXX were 12 different triplet sequences evenly exploring base composition and distribution (see *Supplementary file 3*; for XXX = ACC, template encodes CCC-ACC-UCC to avoid a terminal run of Gs).

Each primer/template pair (0.45/0.525 pmol per reaction) was annealed in 4 mM MgCl$_2$, 1 mM tris•HCl pH 7.4 (80°C 2 min, ramped to 4°C at 0.1°C/s, then kept on ice). The 12 pairs were combined in 0.27M MgCl$_2$/67 mM tris•HCl pH 8.3 on ice to discourage primer-template assortment (of which sequencing later revealed negligible levels). 36 pmol of each triplet in $^{PPP}$NNN (equivalent to 5 µM final concentration after considering eutectic phase equilibration effects upon this more dilute reaction) were added to a reaction vessel in 10.8 µl water, to which 5.4 µl of the primer/template/buffer mix was added followed by 7.2 pmol of ribozyme pre-annealed (80°C 2 min 17°C 10 min, ice >5 min) in 1.8 µl water (f.c. equivalent 1 µM, in excess over the 0.875 µM template to which some ribozymes could tether to enhance extension). Reactions were frozen and incubated (7 days at −7°C) as described above.

Reactions were stopped with 3.6 µl 0.44 M EDTA and 10.5 pmol of each template's competing oligonucleotide (migrating above product, with marker mutations to ensure exclusion), denatured with 6 M urea, and urea-PAGE separated. After alignment with a fluorescence scan of the gel, a region of the sample lane corresponding to primers extended by +4 to +14 nt was excised (encompassing 2–4 triplet additions), and extension products were eluted, precipitated in 77% ethanol with 1 µl 1% glycogen carrier, washed in 85% ethanol and resuspended in water.

These extension products were 3′ adaptor ligated as for fragment sequencing. Products were reverse transcribed (0.2 × adaptor ligation reaction, 1 µM HDVrec primer in Superscript III reaction, 50°C 30 min) and then PCR-amplified (1/30$^{th}$ reverse transcription mix, 0.8 µM each of primers P3HDV and P5GGGX) for sequencing as above (yielding $2 \times 10^5 – 4 \times 10^6$ sequences per ribozyme assay).

After processing and 3′ adaptor trimming, sequences corresponding to primer extended by CCC +1–3 additional triplets were collated for analysis. Variations in upstream primer sequences (see *Supplementary file 3*) allowed the partner template to be identified for each sequenced product; the triplet incorporated after the first CCC was counted. Separately, 10 µl extensions by t5$^{+1}$ of each primer/template alone with its encoded triplet and $^{PPP}$CCC (and $^{PPP}$UCC for the ACC pair) were combined for purification and sequencing as above, to allow isolation of the ribozyme-mediated errors resulting from inclusion of the other 62 (61 for ACC) triplets in the reaction (versus errors from sequencing, recombination etc.). The counts of cognate triplet (C) and each error triplet (E) in the positive control (p) reduced error counts in the experimental samples (x) to yield ribozyme-mediated error counts (E$_r$) thusly: E$_r$ = E$_x$ - E$_p$*(C$_x$/C$_p$) (not reducing E$_x$ below 0, and reallocating all reductions to C$_r$; $^{PPP}$CCC counts (and $^{PPP}$UCC for the ACC template) remained uncorrected).

For each template, counts were then collated at the first/second/third positions to yield base-specific mutation rates for each position (*Figure 8—figure supplement 2*, *Figure 8—source data 1*). Across the 12 triplets, A, C, G, and U were encoded at each position three times; linear averages were calculated to map the position's error profile (*Figure 8b*) and geometric means of the four nucleobases yielded the position's overall fidelity (*Figure 8a*, *Figure 8—figure supplement 1*).

## Triphosphorylated triplet synthesis

Triplets (and some other short oligonucleotides) were prepared from NTPs by T7 RNA polymerase run-off transcription of a 5′ single-stranded DNA overhang downstream of a DNA duplex T7 promoter sequence. In most cases, the 5′ overhang encoded (was the reverse complement of) the desired oligonucleotide. These oligonucleotides were short enough to synthesise during the abortive initiation stage of transcription, attenuating sequence constraints on the first bases of the transcript. However, T7 RNA polymerase exhibited tendencies to skip the first (or even second) base (most

severe for U > C > A > G before second position purines: encoding CGU yielded some $^{PPP}$GU, encoding UAC yielded just $^{PPP}$AC) or use oligonucleotides generated during transcription to re-initiate (e.g. encoding GAG yielded $^{PPP}$GAGAG, encoding AAA yielded $^{PPP}$A$_{6-9}$, encoding UCC yielded $^{PPP}$CCC, encoding CGC yielded some $^{PPP}$GCGC; this tendency was most severe when the oligonucleotide could be accommodated opposite the final template bases of the promoter).

These tendencies could be subverted by encoding additional first bases (usually without providing the corresponding NTP). This initiated the oligonucleotide at the second position where skipping tendencies were lower (e.g. encoding CUAG without CTP yielded $^{PPP}$UAG, encoding UUAC yielded some $^{PPP}$UAC), and reduced recruitment as initiators of products with bases not complementary to the introduced first position template base (e.g. encoding CGAG without CTP yielded $^{PPP}$GAG, encoding CAA without CTP yielded $^{PPP}$AA and $^{PPP}$AAA, encoding AUCC without ATP yielded $^{PPP}$UCC, encoding UCGC without UTP yielded $^{PPP}$CGC).

Each 30 µl transcription reaction contained 72 nmol of each desired product base as an NTP (Roche) (e.g. for $^{PPP}$UCC, 72 nmol UTP, 144 nmol CTP) in 1 × MegaShortScript kit buffer with 1.5 µl MegaShortScript T7 enzyme (ThermoFisher). Also present were 15 pmol of each DNA oligonucleotide forming the transcription duplex target (see *Supplementary file 2*). The reactions were incubated overnight at 37°C, stopped with 3 µl 0.44 M EDTA and 17 µl 10 M urea, and separated by electrophoresis (35 W, 4.5 hr) on a 35 × 18 × 0.15 cm 30% 19:1 acrylamide:bis-acrylamide 3 M urea tris-borate gel. Products were identified through their relative migrations (reflecting overall composition, fastest to slowest: C > U ≈ A > G) by UV shadowing. Triplet bands were excised and eluted overnight in 10 mM tris•HCl pH 7.4, and filtrate (Spin-X) precipitated with 0.3 M sodium acetate pH 5.5 in 85% ethanol. Pellets were washed in 85% ethanol, resuspended in water, and UV absorbances measured with a Nanodrop ND-1000 spectrophotometer (ThermoFisher). Oligocalc (*Kibbe, 2007*) was used to calculate sequence-specific concentrations and yields. $^{PPP}$NNN was generated by combination of equal amounts of each of the 64 triplet stocks in a lo-bind microcentrifuge tube (Eppendorf (Hamburg, Germany)).

3'-deoxy triphosphorylated 'terminator' triplets were transcribed as above but using a 3' deoxynucleoside 5' triphosphate (Trilink biotechnologies) for the last position, migrating faster during PAGE than the equivalent all-RNA triplet. Triplets with 2-thiouridine residues were transcribed as for their corresponding U, replacing UTP with U$^{2S}$TP (Jena Bioscience (Jena, Germany)); incorporation and migration were similar between the two, and their concentrations were calculated from A$_{260nm}$ by comparison to the A$_{260nm}$ of mixtures of the component ribonucleotides with UTP vs. U$^{2S}$TP. Triplets with 2'-fluoro, 2'-deoxy positions could also be transcribed, with lower efficiency, by substituting the corresponding triphosphate (Trilink Biotechnologies). The biotinylated $^{PPP}$GAU–Bio triplet used in γ segment synthesis (*Figure 7c*) was transcribed as for $^{PPP}$GAU, replacing UTP with biotin-16-aminoallyluridine-5'-triphosphate (Trilink Biotechnologies (San Diego, USA)), quantified via by comparison to the A$_{290nm}$ of mixtures of the component ribonucleotides.

Longer triphosphorylated oligonucleotides used in ribozyme self-synthesis were generated similarly, but using ~200 ng of fully double stranded DNA as a template. Candidate product bands were purified and the desired oligonucleotide identified by ribozyme-catalysed in-frame incorporation and, for some, fragment sequencing.

## RNA oligonucleotide/ribozyme preparation

Transcriptions were performed on ~15 ng/µl dsDNA using MegaShortScript enzyme and buffer (ThermoFisher) with 7.8 mM of each NTP, or, to yield a 5' monophosphate on the product to avoid aberrant ligation, 10 mM GMP (guanosine monophosphate) and 2 mM of each NTP ('GMP transcription').

dsDNA templates for some of these (in *Supplementary file 3*) were generated ('fill-in') using three cycles of mutual extension (GoTaq HotStart, Promega) between the associated DNA oligonucleotide and 5T7 (or, where indicated, HDVrt for defined 3' terminus formation [*Schürer et al., 2002*]) followed by column purification (QiaQuick, Qiagen).

Some 5' biotinylated RNAs were synthesized using the TGK polymerase (*Cozens et al., 2012*) (56 µg/ml, in 1 × Thermopol buffer (NEB) supplemented with 3 mM MgCl$_2$) to extend 5' biotinylated RNA primers (0.75 µM) on DNA templates (1 µM) using 2.5 mM of each NTP (94°C 30 s, 45°C 2 min, 65°C 30 min, 45°C 2 min, 65°C 30 min, then all repeated). Biotinylated products were bead-purified as above.

3' biotinylation of RNAs was achieved in two stages: 3' azidylation (at 2 µM with 25 U/µl yeast poly-A polymerase (ThermoFisher) and 0.5 mM 2'-azido-2'-deoxycytidine triphosphate (Trilink Biotechnologies) for 1 hr at 37°C) with subsequent acidic phenol/chloroform extraction and 75% ethanol precipitation, then copper-catalysed biotin-(PEG)$_4$-alkyne (ThermoFisher) cycloaddition (*Winz et al., 2012*) with subsequent 75% ethanol precipitation followed by resuspension and buffer exchange in Ultracel 3K filters (Amicon) to remove residual biotin-alkyne.

## Selection library synthesis

Round one libraries were synthesised by mutual extension of 4 nmol of oligonucleotides 1baN30 and 1GMPfo or 1GTPfo at 1 µM each in 1 × isothermal amplification buffer (NEB) with 250 µM each dNTP, annealed (80°C 3 min, 65°C 5 min) before addition of 0.4 U/µl Bst 2.0 (NEB) and 30 min 65°C incubation.

After purification, 375 µg of each DNA (~$1.5 \times 10^{15}$ molecules) were transcribed in 5 ml transcription reactions (36 mM tris•HCl pH 7.9 (at 25°C), 1.8 mM spermidine, 9 mM DTT, 10.8 mM MgCl$_2$, 2 mM each NTP, 1% 10 × MegaShortScript buffer, 2% 1:9 MegaShortScript:NEB T7 RNA polymerase, 37°C overnight). These were treated with DNase, acid phenol/chloroform extracted and 73% ethanol precipitated prior to urea-PAGE purification, elution, filtering (Spin-X) and re-precipitation, yielding the 1GTP Zcore selection construct (*Supplementary file 1*). 10 mM GMP was present in transcriptions of the 1GMP construct, and for future transcriptions of the GMP construct selection branch and rounds 8–18; round 19–21 and reselection libraries were transcribed without GMP. Most subsequent selection rounds were transcribed in 1/10$^{th}$ scale transcriptions with 15 µg of DNA (~$6 \times 10^{13}$ molecules) derived from amplification of recovered PCR products (see later).

For round 8, 700 pmol DNA was formed, with Tri3CUUQ amplifying round seven merged output (50 pmol), round seven merged output recombined by StEP (*Zhao and Zha, 2006*) (200 pmol), and 0core ribozyme with the starting 3' N$_{30}$ library domain added (450 pmol, but extinct at the end of selection). DNA encoding type 5$^s$ amplified with AACAt5s was used to generate reselection libraries by PCR amplification using primers TriGAA7GAAM and T5ba13N/T5ba20N/T5ba28N; 5 pmol of the three dsDNA products were transcribed to generate reselection constructs.

## In vitro evolution cycle

An outline of the selection strategy is shown in *Figure 1—figure supplement 2*, with detailed lists of selection oligonucleotides and extension parameters in *Figure 1—source data 1*, *Figure 2—source data 1*, *Figure 4—source data 1* and *Supplementary file 3*. First, selection construct was annealed with equimolar dual-5' biotinylated primer in water (80°C 2–4 min, 17°C 10 min), then chilled extension buffer and triplets were added before freezing and −7°C incubation.

At the end of incubation the reaction was thawed on ice. To link the primer 3' hydroxyl to the 5' monophosphate of GMP constructs, selection constructs were buffer-exchanged directly after thawing using a PD-10 column (GE) in a cold room, into 3 ml ligation mix (optimised to prevent ligation over gaps) (2 mM MgCl$_2$, 50 mM tris•HCl pH 7.4, 0.1 mM ATP, 1 mM DTT, 2 µM $^{HO}$GCG (Rounds 1–7) or 2 µM $^{HO}$CUG (Rounds 8–18) with 30 U/ml T4 RNA Ligase 2 (NEB)). After incubation at 4°C for 1 hr, these were stopped with 2.2 mM EDTA and acid phenol/chloroform treated.

Constructs were then precipitated with glycogen carrier and 0.3 M sodium acetate in isopropanol (55%) before resuspension and denaturation (94°C 4 min, in 6M urea 10 mM EDTA with a 3 × excess of competing oligonucleotide against the primer). For the reselection rounds, constructs were then treated with polynucleotide kinase (NEB) before denaturation to resolve the HDV-derived 2', 3'-cyclic phosphates and allow later adaptor ligation.

Constructs were urea-PAGE separated alongside FITC-labelled RNA markers equivalent to successfully ligated constructs. The marker-adjacent gel region in the construct lane was excised, excluding the bulk unreacted construct (judged by UV shadowing). Biotinylated (primer-linked) constructs were eluted overnight into BB with 100 µg MyOne C1 beads. After 30 µm filtering (Partec Celltrics(Wolflabs (York, UK))) of the supernatant to remove gel fragments, the beads were washed in BB then denaturing buffer (8 M urea, 50 mM tris•HCl pH 7.4, 1 mM EDTA, 0.1% Tween-20, 10 µM competing oligonucleotide, 60°C 2 min) to confirm covalent linkage of construct to primer, before further BB washing and transfer (to a fresh microcentrifuge tube to minimize downstream contamination). At this stage in the reselection, 3' adaptors were then ligated to bead-bound constructs as

above for 2 hr (with buffer/enzyme added after bead resuspension in other reaction components including 0.04% Tween-20), and beads BB washed and transferred again.

Bead-bound constructs were now reverse-transcribed using 1 μM RTri (or HDVRec for the reselection) by resuspension in a Superscript III reaction with added 0.02% Tween-20 (50°C 30 min). Beads were BB washed and the RNA-bound cDNA 3' end blocked by incubation with terminal deoxynucleotidyl transferase (ThermoFisher) and 0.2 mM dideoxy-ATP (TriLink) with 0.02% added Tween-20 (37°C 30 min), and beads were BB washed and transferred again.

cDNAs were eluted (10 μl 0.1 M NaOH 0.1% Tween-20 20 min), neutralized and plus strands regenerated with 0.2 μM rescue oligonucleotide in an IsoAmp II universal tHDA kit (NEB) reaction (65°C 60 min) to read through the structured product region. Whilst at this temperature, reactions were stopped with 5 mM EDTA and one volume of BB with 50–100 μg beads to bind the nascent biotinylated plus strands at room temperature. These beads were then BB washed, NaOH washed again to discard cDNAs (and recover only correctly-primed plus strands), and washed and transferred again.

Each 50 μg of beads were then subjected to plus strand recovery PCR in a 100 μl GoTaq HotStart reaction with 0.5 μM each RTri (or HDVrec for reselection) and RecInt (rounds 1–5)/RecIntQ (rounds 6–9)/RecIntL (rounds 10–14, 19–21 and reselection)/RecIntQL (rounds 15–18). The product was agarose size-purified, $A_{260nm}$ quantified and added to construct synthesis PCR in 3 × molar amount of the anticipated recovered RNA (judged by test extensions) that yielded it.

This final PCR for construct transcription in the subsequent selection round used 1 μM of the indicated construct synthesis primer, plus 1 μM RTri (or HDVrt for the reselection), in GoTaq HotStart reactions or (where indicated in source data) the GeneMorph II kit for mutagenesis (Agilent (Santa Clara, USA)).

Conditions for selections are included as source data 1 for *Figures 1*, *2* and *4*. Numerical data for *Figures 1c* and *3b* are included in *Figure 1—source data 2* and *Figure 3—source data 1* respectively. Numerical data and calculations for *Figure 8* and its *Figure 8—figure supplement 1* and *2* are supplied as *Figure 8—source data 1*. Numerical data and calculations for *Figure 8—figure supplement 3* are supplied as *Figure 8—source data 2*. A more extensive selection of substrate concentration-dependent error rates is supplied in *Figure 9—source data 1*. Sequences of ribozymes, triplet synthesis templates, and oligonucleotides used in this study are supplied in *Supplementary files 1*, *2* and *3* respectively.

## Acknowledgements

We thank S Thomsen, M McKie and M Sharrock for important discussions, and BT Porebski and SL James for assistance with sequencing analysis. This work was supported by a Homerton College junior research fellowship (JA.), a Harvard-Cambridge summer fellowship (AR.) and by the Medical Research Council (JA., EG., AM., PH.; programme no. MC_U105178804).

## Additional information

### Funding

| Funder | Grant reference number | Author |
| --- | --- | --- |
| Medical Research Council | MC_U105178804 | Philipp Holliger |

The funders had no role in study design, data collection and interpretation, or the decision to submit the work for publication.

### Author contributions

James Attwater, Conceptualization, Formal analysis, Investigation, Methodology, Writing—original draft; Aditya Raguram, Acquired and analysed minor groove modification effect data; Alexey S Morgunov, Characterised initial Zcore activity; Edoardo Gianni, Analysed ribozyme heterodimer formation by gel mobility shift; Philipp Holliger, Conceptualization, Supervision, Methodology, Writing—original draft

Author ORCIDs
James Attwater (iD) http://orcid.org/0000-0002-7244-9910
Philipp Holliger (iD) http://orcid.org/0000-0002-3440-9854

Decision letter and Author response
Decision letter https://doi.org/10.7554/eLife.35255.044
Author response https://doi.org/10.7554/eLife.35255.045

## Additional files

### Supplementary files

• Supplementary file 1. Ribozyme sequences. All sequences are written in a 5′-to-3′ direction, and generated by GMP transcription of the corresponding PCR-generated dsDNA of the sequence downstream of 5T7 sequence duplex. All transcripts were PAGE-purified. HDV ribozyme sequences (blue) were transcribed in series with reselected type 5 ribozymes and cleave themselves off during transcription (*Schürer et al., 2002*) to yield precise 3′ ends with 2′, 3′-cyclic phosphates; the presence of this group did not affect type 5 activity. Sequences corresponding to the 5′ 'cap+' or 'cap−' regions from the selection, presenting a target for type 1 interaction, are coloured light green, with alternative arbitrary inert 5′ hairpin-forming sequences in dark green. Single-stranded sequences capable of hybridization with sites at the 3′ (or 5′, type 5$^{cis}$) ends of certain templates to endow flexible ribozyme-template duplex tethering (*Wochner et al., 2011*; *Attwater et al., 2010*) are coloured yellow. Accessory domains 3′ of the catalytic core are in bold.
DOI: https://doi.org/10.7554/eLife.35255.039

• Supplementary file 2. Transcription of triplets. Each target was transcribed from oligonucleotides 5T7 (*Supplementary file 3*) and 5′-(var)-TATAGTGAGTCGTATTAATTTCGCGGGCGAGATCGATC-3′, where the (var) overhang encodes (is the DNA reverse complement of) the sequence indicated below. Guide yields: >15 nmol = ***, 10–15 nmol = **, 5–10 nmol = *, <5 nmol = ~. †: These transcriptions yield one main product suitable for use as markers to identify comigrating triplets with a similar G/AU/C content.
DOI: https://doi.org/10.7554/eLife.35255.040

• Supplementary file 3. Oligonucleotide sequences. All sequences are written in a 5′-to-3′ direction. DNA sequences are coloured grey. RNA sequences are coloured black, with the exception of hammerhead ribozyme (brown) and HDV ribozyme (blue) sequences transcribed in series that cleave themselves off during transcription (*Schürer et al., 2002*) to yield precise 5′ and 3′ ends respectively. All RNAs were denaturing PAGE-purified, and DNAs were not, unless otherwise noted ('(non)-GP'). Competing oligonucleotides, listed beneath templates, were purified using RNeasy columns (Qiagen). Primer and oligonucleotide binding sites on templates are underlined.
DOI: https://doi.org/10.7554/eLife.35255.041

• Transparent reporting form
DOI: https://doi.org/10.7554/eLife.35255.042

### Data availability

All data generated during this study and collated sequencing results are included in the manuscript and its supporting files. Source data files have been provided for Figures 1-4, 8 and 9.

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
