## [Decision Letter]

Thank you for submitting your article "Ribozyme-catalysed RNA synthesis using triplet building blocks" for consideration by *eLife*. Your article has been reviewed by three peer reviewers, and the evaluation has been overseen by a Reviewing Editor and James Manley as the Senior Editor. The following individual involved in review of your submission has agreed to reveal his identity: Nebojsa Janjic (Reviewer #3).

The reviewers have discussed the reviews with one another and the Reviewing Editor has drafted this decision to help you prepare a revised submission. The most important thing to improve is the written clarity of the manuscript. There was a lot of agreement between the reviewers' comments so we attach their full reviews below for your attention. All their concerns should be attended to.

*Reviewer #1:*

Attwater et al. report an advance in efforts to develop a robust self-replicating RNA by presenting an example of an RNA polymerase ribozyme (RPR) capable of transcribing RNA from structured RNA templates of substantial length with fairly good fidelity. This RPR uses a trinucleotide building block (in contrast to NTPs used by standard polymerases), that the authors speculate could have implications in the evolution of mRNA decoding by the ribosome.

The major findings from this work are as follows:

1) The authors exploit the ligase activity in the catalytic core of their previously reported RPR Z and evolve a suite of catalytic sequences capable of extending an RNA primer using RNA triplets as substrates. Through further iterations of in-vitro selection for triplet dependent RPRs, an RNA sequence emerged without any catalytic activity itself, but it enhanced triplet extension by these newly evolved RPR sequences (named t2 through t6) through a bimolecular interaction between the catalytically proficient sequence. Further rounds of selection using the cofactor t1 resulted in the most efficient triplet RPR, t5^+1^ which was used for downstream studies. This RPR could use structured hairpins as templates, even those with Tm >90oC that were intractable to previous RPRs that use NTPs as substrates. The authors demonstrate the efficiency of this RPR by transcribing both + and – strands of the Broccoli aptamer, a structured and fluorescent RNA.

2) The authors used t5^+1^ RPR to replicate fragments of itself (both + and – strands), using triplet building blocks (and in some cases hexamers). The complete active RPR could be reconstituted from the five individual components of the RPR transcribed by the same RPR. Although this replicase ribozyme is still not able to replicate itself in its entirely, it does give rise to a functioning replicase by using building blocks as small as trimers, Previous efforts had primarily used the ligase activity of the RPRs to assemble oligonucleotides representing their sequence to assemble a 'replica' of itself. Considering the challenges of self-replication, even incremental advances in the right direction are significant.

Overall this work introduces a new RPR system that brings the field closer to a self-replicating system, which is one of the cornerstones of Origin of life research. The authors explore properties of this replicase including fidelity and catalytic efficiency and explain the molecular basis for these properties. This work also introduces trinucleotides as plausible substrates on which ancient RNA-based replicases could have acted. Considering the relative ease of non-enzymatic polymerization of monomers into triplets, the presence of these triplets in a prebiotic setting seems plausible. The current work is detailed and the authors have explored multiple facets of the new system they have setup. This new RPR presents a major advance in terms of its ability to create a fully functional copy of itself (albeit in fragments). As such, this would be a great addition to the current understanding of RNA-based self-replication systems.

1) Overall, the manuscript was very difficult to follow. It is up to the authors to make their presentation as simple and direct as possible. Given the availability of supporting information, there is no reason why the authors could not add schemes where appropriate to clarify their presentation, logic, and workflow.

2) Regarding the dimerization, it is not clear how the authors implicated the 5'-hairpin to begin with, leading to the experiment that showing that a single mutation in that loop inhibits dimerization.

3) The assertion that the two RNAs form a 1:1 complex is not quantitatively substantiated from Figure 3B.

4) Considering their success in replicating structured hairpins and achieving primer free synthesis of an 18 nt strand (β+), it would be interesting (though not necessary for publication) to see how much of the Broccoli RNA could be transcribed without the addition of a primer sequence. If this is not possible, the authors could demonstrate the complete replication of a functional RNA of tractable length, like the Hammerhead ribozyme, to illustrate the ability of their trinucleotide RPR to replicate entire sequences of functional RNAs by exclusively using triplets as substrates.

5) The fidelity analysis is based on testing 12 of the possible 64 trinucleotides, so one cannot be convinced of the generality of their observations and inferences. The authors need to underscore this in the discussion of fidelity. The authors show that increased concentrations of triplets help fidelity, however in the absence of the 'correct' triplet, it is likely that RPR will incorporate 'mismatch' triplets. The authors should comment on this aspect.

*Reviewer #2:*

1) In this paper, the authors demonstrate interesting and novel results that provide further evidence for RNA's capacity to act as prebiotic genetic material. The key feature associated with this study involved using trinucleotide triphosphates (triplets) rather than NTPs as substrates for a novel RNA polymerase ribozyme. The use of triplets allowed structured templates to be unfolded and in some cases avoided the need for primers, circumventing an ongoing obstacle to RNA self-replication. This study thus makes some significant steps towards identifying a truly self-replicating prebiotic biopolymer. Literature precedents are well-discussed, and the results are interpreted in a detailed and convincing manner. I would nonetheless encourage the authors to try to make these important results as accessible as possible to readers who are unfamiliar with in vitro evolution.

Specific comments:

2) Subsection “in vitro evolution of triplet polymerase activity”, last paragraph. The use of "in-ice evolution" (a technique developed by the authors) is reported. A sentence could be added describing why this technique is being used / why it is necessary.

3) For certain experiments (e.g. subsection “Fidelity of triplet-based RNA synthesis”), random triplet pools were used. What kind of triplet pools were used in other sequence-copying experiments? Were they non-random? This needs to be more clearly articulated.

4) "… we found that using modified substrates with a disrupted minor groove hydrogen bond acceptor at the 3rd position…" – What is the modification? This should be stated in the body-text.

5) What is the scope of substrates for the t5^+1^ ribozyme? Triplets reportedly incur a lower entropic cost than NTPs (Discussion, third paragraph), but what would happen when a mixture of the two were used? Would both the NTPs and triplets be incorporated?

If these points are satisfactorily addressed, then I am in favour of acceptance.

*Reviewer #3:*

Attwater et al. used an in vitro evolution method to enhance the trace amount of ligase activity with short (3-nucleotide) substrates inherent in the truncated version of the Z RPR ribozyme (called "Zcore"). This was achieved by adding a random region of 30 nucleotides at the 3' end of Zcore and a template at the 5' end, followed by selection for variants able to extend a primer hybridized to the 5' template with two 5'-triphosphate triplets and ultimately ligate the extended primer to the 5' terminal triphosphate. After 7 rounds of activity enrichment, the authors identified a sequence that represented 25% of the activity-enriched pool called "type 0" that, upon truncation to its minimal active sequence ("0core"), was able to catalyze multiple cycles of templated ligation with triplet substrates, including the ability to extend thought double-stranded regions of the template by cooperative invasion through the secondary structure of the template (this was not achievable with monomer substrates, NTPs).

After additional 14 rounds selection, the composition of the pool showed considerable changes: type 0 sequence had disappeared and was replaced by six new types of sequences, types 1-7. Remarkably, the most prevalent type 1 sequence (50% of the pool) was completely inactive and types 2-6 were less active than the activity enriched pool. This mystery was solved by the observation that a truncated version of the type 1 sequence, while being inactive, can form augment the activity of type 2-6 ribozymes by forming heterodimeric complexes. In the presence of type 1 sequence, the best stand-alone ribozyme, type 5, no longer required the template to be tethered to the 5' end of the type 5 ribosome, in a remarkable example of an intermolecular complex between the dimeric type 1:type5 ribozyme, the untethered primer template, and the triplet substrates. Further optimization of the type 5 ribozyme by re-randomization of the previously fixed 3' end sequences and reselection in the presence of minimal type 1 sequence (called "1") resulted in the final heterodimeric holoenzyme called "t5^+1^. t5^+1^ was able to catalyze the extension through templates with Tm values of 93 ⁰C, and was capable of generating both the functionally active Broccoli RNA aptamer and its complementary template. Finally, t5^+1^ was able to replicate its own t5 sequence split into five segments, mostly from triplets, with some help from longer substrates (hexamers) needed for two of the segments, thus falling short of having the capacity for a full replication cycle, but nonetheless, exhibiting impressive catalytic efficiency.

The surprising observation that t5^+1^ can extend primers in both the canonical 5'-3' as well as the reverse 3'-5' direction raises the potential for primer extension from short sequences in a primer-free manner, therefore potentially solving the "primer problem" (depletion of longer but presumably sparse primers needed for replication in prebiotic conditions). Impressive overall positional fidelity of more than 97% was achieved by counter-selection for mispairing triplets with a clever inclusion of excess 3'-deoxy terminator triplets during late rounds of selection. Attwater et al. mapped the region responsible for fidelity to the 3' domain of t5. Further reduction in mis-incorporation comes from cognate triplet-triplet interactions have a self-correcting feature of countering the effect of triplets with strong base-pairing tendency.

The authors make a strong case that triplets represent relatively short and yet powerful substrates able to invade highly stable structural features in RNA templates. The evolution of a heterodimeric ribozyme capable of generating functional RNAs with triplet substrates represents a major accomplishment that advances the plausibility of RNA-catalyzed RNA replication in prebiotic environment. Despite its considerable length, as well as its remarkable breadth and scope, the manuscript reads like a riveting novel. It is a tour de force scientific accomplishment, and as such, it will be of considerable interested to a wide audience of readers of *eLife*. [Editors’ note: this reviewer’s minor comments have not been included.]

---

## [Author Response]

Reviewer #1:

[…] 1) Overall, the manuscript was very difficult to follow. It is up to the authors to make their presentation as simple and direct as possible. Given the availability of supporting information, there is no reason why the authors could not add schemes where appropriate to clarify their presentation, logic, and workflow.

We apologize for the poor presentation and have made several changes to enhance clarity. These include textual changes to reduce ambiguity and lay out our reasoning more thoroughly as well as explanatory schemes of ribozyme activity in Figure 1 and Figure 1—figure supplement 1. Furthermore, we have redesigned Figures 3 and 7, and now provide an additional dataset presenting substrate concentration-dependent error rates in a more straightforward manner in Figure 9—figure supplement 1.

The manuscript structure is complicated by the multiple stages of RNA evolution and characterization involved in the project, and to help navigate the progress of our ribozyme development, we now provide the new Figure 4—figure supplement 1 detailing all the stages of evolution and engineering of the different triplet polymerase ribozymes to provide a “roadmap”.

2) Regarding the dimerization, it is not clear how the authors implicated the 5'-hairpin to begin with, leading to the experiment that showing that a single mutation in that loop inhibits dimerization.

We welcome the opportunity to clarify this. We had originally noticed the ‘cap-’ mutation in the most abundant isolated type 1 clones, and when testing this in different constructs observed that it quenched type 1 enhancement pointing towards the 5’ hairpin region as the critical interaction site. We now refer to this in the manuscript text.

Our knowledge about this dimerization interface is currently limited and we have therefore generally refrained from speculation. However, it seems possible that the ‘cap-’ mutation was enriched in these type 1 sequences to diminish homodimerisation of type 1 that may interfere with heterodimerisation with active triplet polymerase in the selection pool.

3) The assertion that the two RNAs form a 1:1 complex is not quantitatively substantiated from Figure 3B.

We now include quantitative data on the stoichiometry of the complex as observed by gel shift (EMSA) titration in Figure 3B, and activity tests in Figure 3—figure supplement 1. These support a 1:1 stoichiometric complex in the active ribozyme as expected from the final selection pool composition.

4) Considering their success in replicating structured hairpins and achieving primer free synthesis of an 18 nt strand (β+), it would be interesting (though not necessary for publication) to see how much of the Broccoli RNA could be transcribed without the addition of a primer sequence. If this is not possible, the authors could demonstrate the complete replication of a functional RNA of tractable length, like the Hammerhead ribozyme, to illustrate the ability of their trinucleotide RPR to replicate entire sequences of functional RNAs by exclusively using triplets as substrates.

The reviewer highlights an important challenge that such ribozymes must ultimately face – copying without primers both strands of functional RNAs to achieve their complete replication using only short oligonucleotide substrates. With the current triplet polymerase ribozyme, while primer-initiated synthesis is almost general (as we show by synthesizing both + and – strands of the Broccoli RNA aptamer), we cannot currently achieve primer-free synthesis of all sequences. The latter includes Broccoli where yields of full-length RNA are poor using triplets alone. This may reflect inefficient initiation of synthesis or poor upstream inclusion of some 5’ terminal triplets.

Nevertheless, the current triplet polymerase ribozyme can achieve this for some functional sequences, most relevantly from t5 itself. To demonstrate this, we have now included data on the primer-free synthesis and replication of the Υ segment. Specifically, we describe primer-free synthesis using triplets alone of the γ+ strand segment with sufficient yield to use this ribozyme-synthesized RNA as the template for primer-free γ- strand synthesis, demonstrating complete primer-free sequence replication using only triplets. These new data are now included as Figure 7C, replacing the (now in our view redundant) primer-free γ+ syntheses.

5) The fidelity analysis is based on testing 12 of the possible 64 trinucleotides, so one cannot be convinced of the generality of their observations and inferences. The authors need to underscore this in the discussion of fidelity. The authors show that increased concentrations of triplets help fidelity, however in the absence of the 'correct' triplet, it is likely that RPR will incorporate 'mismatch' triplets. The authors should comment on this aspect.

We measured fidelity of triplet incorporation with all64 triplet substrates present but in the context of 12 defined template triplet sequences flanked by conserved CCC incorporation sites. These were chosen to be compositionally balanced, including both GC-rich and AU-rich template triplets, to provide fair estimates of ribozyme fidelity, as well as an effective tool to compare the fidelities of different ribozymes.

This assay was designed to allow us to separate incorporation fidelity (i.e. discrimination among 64 triplets for decoding a defined anti-triplet) from each ribozyme’s extension capabilities by providing a uniform, tractable sequence context.

The important qualification regarding this assay, we feel, is therefore that while it provides an accurate representation of fidelity in a defined sequence context (triplet-triplet junction), it may not provide a comprehensive picture of triplet polymerase fidelity performance in all sequence contexts. We now provide this caveat in the text.

Reviewer #2:

*1) In this paper, the authors demonstrate interesting and novel results that provide further evidence for RNA's capacity to act as prebiotic genetic material. The key feature associated with this study involved using trinucleotide triphosphates (triplets) rather than NTPs as substrates for a novel RNA polymerase ribozyme. The use of triplets allowed structured templates to be unfolded and in some cases avoided the need for primers, circumventing an ongoing obstacle to RNA self-replication. This study thus makes some significant steps towards identifying a truly self-replicating prebiotic biopolymer. Literature precedents are well-discussed, and the results are interpreted in a detailed and convincing manner. I would nonetheless encourage the authors to try to make these important results as accessible as possible to readers who are unfamiliar with* in vitro *evolution.*

As discussed above, we have made a significant number of changes to the manuscript in order to present our results more clearly.

These include textual changes to improve clarity and remove ambiguities. We have also made an effort to lay out our experimental progress and reasoning more thoroughly, using explanatory schemes of ribozyme activity in Figure 1 and Figure 1—figure supplement 1, redesigning Figures 3 and 7, and providing an additional more straightforward dataset in Figure 9—figure supplement 1. We hope this will make the manuscript easier to read and more accessible to the general reader.

Specific comments:2) Subsection “in vitro evolution of triplet polymerase activity”, last paragraph. The use of "in-ice evolution" (a technique developed by the authors) is reported. A sentence could be added describing why this technique is being used / why it is necessary.

We have expanded our description of the mechanisms of in vitro evolution and discuss in more detail the reasoning behind our use of eutectic ice phases and in-ice evolution. As the properties of ice as a medium for RNA-catalyzed RNA synthesis, replication and evolution are discussed at length in the cited references, a fairly brief discussion must suffice. We hope nevertheless that this will now clearly illustrate our rationale in using this technique in our work.

3) For certain experiments (e.g. subsection “Fidelity of triplet-based RNA synthesis”), random triplet pools were used. What kind of triplet pools were used in other sequence-copying experiments? Were they non-random? This needs to be more clearly articulated.

We generally list the triplet substrates used for different experiments but apologise for any ambiguity.

In general for preparative syntheses we have avoided using pools of random triplets, providing instead a set of defined triplets as specified by the template sequence to minimise substrate consumption. We have however verified that random triplet pools can be used for key triplet based RNA synthesis experiments including hairpin invasion (see Figure 5—figure supplement 1) and ribozyme segment synthesis (see Figure 6—figure supplements 4 and 5), as well as in experiments on ribozyme fidelity (see e.g. Figure 8). We now explicitly state the substrate composition in the text and figure legends for all experiments described.

4) "… we found that using modified substrates with a disrupted minor groove hydrogen bond acceptor at the 3rd position…" – What is the modification? This should be stated in the body-text.

We apologize for this omission and have amended this. We now state in the text that uracil was replaced by 2-thio uracil in the triplets.

5) What is the scope of substrates for the t5^+1^ ribozyme? Triplets reportedly incur a lower entropic cost than NTPs (Discussion, third paragraph), but what would happen when a mixture of the two were used? Would both the NTPs and triplets be incorporated?

This is an interesting suggestion considering that abiotically-generated substrate pools would likely include a mixture of different length substrates including mono- and dinucleotides. We have performed additional experiments to explore this and now include data on their incorporation during segment synthesis in Figure 6—figure supplement 5.

In summary, t5^+1^ does incorporate both dinucleotide triphosphates and NTPs, but with reduced efficiency. Furthermore, they appear to serve as poor substrates for segment synthesis with triplets incorporated much more efficiently. Increasing amounts of NTP and dinucleotide substrates vs. triplets (up to a total 4:2:1 ratio) do lead to diversification of extension products outside the initial triplet register, but do not notably affect the total amount of ligation performed.

As seen in Figure 6—figure supplement 2, more length-heterogenous substrate pools (including longer oligomers) may offer both advantages and disadvantages, but we feel that a more systematic investigation of diverse substrate pools and their effect on yield and fidelity of RNA-catalyzed RNA replication goes beyond the scope of the current manuscript.